# Oolong: Evaluating Long Context Reasoning and Aggregation Capabilities

## Abstract

As model context lengths continue to grow, concerns about whether models effectively use the full context length have persisted. While several carefully designed long-context evaluations have recently been released, these evaluations tend to rely on retrieval from one or more sections of the context, which allows nearly all of the context tokens to be disregarded as noise. This represents only one type of task that might be performed with long context. We introduce Oolong, a benchmark of long-context reasoning tasks that require analyzing individual chunks of text on an atomic level, and then aggregating these analyses to answer distributional questions. Oolong is separated into two task sets: Oolong-synth, a set of naturalistic synthetic tasks, where we can easily ablate components of the reasoning problem; and Oolong-real, a downstream setting which requires reasoning over real-world conversational data. Oolong requires models to reason over large quantities of examples, to perform both classification and counting in-context, and to reason over temporal and user relations. Even frontier models struggle on Oolong, with GPT-5, Claude-Sonnet-4, and Gemini-2.5-Pro all achieving less than 50% accuracy on both splits at 128K. We release the data and evaluation harness for Oolong to enable further development of models that can reason over large quantities of text.

## 1 Introduction

In the last several years, the exponentially increasing context lengths of LLMs have enabled many new applications, including reasoning models (Guo et al., 2025), many-shot prompting (Bertsch et al., 2024), and repository-level code generation (Jimenez et al., 2023). However, despite the rapid improvement in this area, long-context aggregation remains a challenging capability to study. Tasks like summarization are realistic but produce long and subjective outputs that are challenging to evaluate. Other tasks require only a subset of the input, which could be extracted from the long context before solving (e.g. using a RAG system).

Multi-step reasoning datasets make progress towards measuring this capability, but often lack realism – inputs may be artificial (e.g. random repeated words in RULER (Hsieh et al., 2024)), the posed questions may be unrealistic (e.g. "identify the second poem about this topic" in MRCR (OpenAI, 2025)), and the relevant context may be embedded in distractor text from a different domain or a well-known text (e.g. small state updates embedded in Project Gutenberg novels in BABILong (Kuratov et al., 2024)). We argue that aggregation is best measured in realistic settings that require multiple *types* of aggregative component tasks– including the ability to draw inferences across different sections of input and perform basic numeracy operations with this information (e.g. counting, measuring relative frequency).

We propose Oolong, a benchmark that requires multi-hop reasoning over long inputs to produce easily verifiable outputs. Oolong is separated into two task sets, presented in Figure 1. Oolong-synth (§2) is a set of naturalistic synthetic tasks constructed from existing in-context learning datasets. These tasks require implicitly labeling the examples in-context to reason over distributional properties of the labels (*counting tasks*), over user-specific patterns (*user tasks*), and over changes in the label distribution over time (*temporal tasks*). Oolong-real (§3) poses the same types of questions over real data that is not so easily separable into component parts– asking challenging

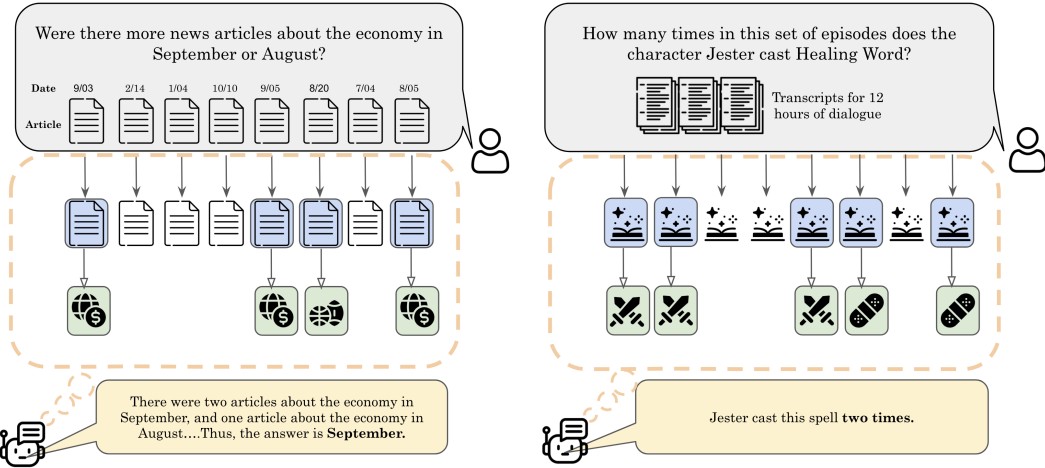

Figure 1: OOLONG poses questions that require performing a multi-step information aggregation process to determine the solution. OOLONG-synth uses ICL-based tasks, which could be easily decomposed and solved iteratively, as a proxy for real-world aggregation tasks over long inputs. OOLONG-real poses challenging information aggregation questions over transcripts from live-action Dungeons & Dragons shows, which can not be easily decomposed into component pieces.

questions about character states and campaign statistics from live-play Dungeons & Dragons role-playing transcripts, using human-annotated gold answers.

Each task requires identifying the relevant segments of the input; this ranges from only a handful of instances or lines of dialogue to questions that require the use of *every* line of the input. The relevant segments must be classified or categorized, and these individual decisions must be aggregated to produce a final answer. By framing the benchmark around problems that are *simple individually*, we ensure that we are measuring capability on long context reasoning and not accuracy on the underlying task. By requiring identification of relevant context, classification decisions over that context, and numeracy skills to produce statistics about the input, OOLONG requires models to perform multi-step reasoning over long inputs.

Although these component capabilities have been well studied and the frontier models demonstrate strong performance in each (Yen et al. (2024), Agarwal et al. (2024), *inter alia*), we find that models struggle with information aggregation as the input length grows. None of the models benchmarked score higher than 50% on OOLONG at 128K context. We study why this task is so challenging using the more flexible OOLONG-synth, ablating settings that decrease context length dramatically and remove the line-by-line classification task, and find that identification and aggregation of information is the bottleneck, not labeling (§4). We also study the role of reasoning behavior and identify consistent error cases in reasoning models that limit their performance in this type of task (§4). We believe OOLONG is a usefully challenging evaluation of long-context reasoning abilities.

## 2 OOLONG-SYNTH

We aim to construct a synthetic information aggregation task that allows fine-grained control over the types of information and number of steps necessary to solve the task. To do this, we need building blocks: simple, short-context documents to build into a longer collection. We construct OOLONG-synth by constructing challenging corpus-level questions over existing in-context learning (ICL) datasets.

### 2.1 DATA

We collect 10 common text classification datasets with between 2 and 10 labels. We select for tasks that are possible for the authors to perform without difficulty (and validate that these tasks are similarly simple for models during our dataset filtering). Table 1 describes the datasets in more

| Dataset | Task | # Labels | Input Len |
|---|---|---|---|
| Spam (Almeida et al., 2011) | SMS spam classification | 2 | 57 |
| TREC-QC-coarse (Li & Roth, 2002; Hovy et al., 2001) | Question type classification | 6 | 39 |
| AGNews (Zhang et al., 2015) | Headline topic classification | 4 | 90 |
| App Reviews (Zur, 2017) | Review sentiment classification | 2† | 49 |
| Pavlick Formality (Lahiri, 2015; Pavlick & Tetreault, 2016) | Formality classification | 2† | 51 |
| IMDB reviews (Maas et al., 2011) | Sentiment analysis | 2† | 376 |
| HiTZ Negation (García-Ferrero et al., 2023) | Verify claims about definitions | 2 | 45 |
| Yahoo Topics (Zhang et al., 2015) | Question topic classification | 10 | 74 |
| MultiNLI (Williams et al., 2018) | Entailment | 3 | 70 |
| Metaphors (Bizzoni & Lappin, 2018) | Metaphor meaning validation | 2 | 51 |

Table 1: Datasets used for OOLONG-synth. † indicates cases where we combined similar labels to produce a smaller label-set for OOLONG. The input length is the average per-instance length (in Llama 2 tokens), including our added date and user metadata.

detail. We split the data into two validation tasks and eight test tasks. The validation tasks are selected to avoid underlying task overlap with test tasks (in the style of Min et al. (2022)), and so that there is both a 2-label and a multi-label (6-label) validation task. For the sentiment and formality datasets with more than two labels, we reduce the label space to a single positive and a single negative label, to reduce the difficulty of the task.

**Data filtering** In a normal in-context learning task, if a few examples are unusually hard or even mislabeled, this has a small impact on the overall score. Because we intend to require aggregation across many ICL examples at once, one particularly difficult example can affect the score on many questions downstream. However, many ICL datasets are known to contain mislabelings (Ying & Thomas, 2022; Chong et al., 2022; Klie et al., 2023).

We perform an additional screening step to remove mislabeled or unusually hard examples. We select two models that we do not expect to be substantially stronger than the models we are evaluating: GPT-4.1 nano and Llama 4 Maverick. We perform zero-shot ICL with a minimal instruction that provides the label space and type of task. We then exclude all examples that *both* models get incorrect, for exclusion rates ranging from 0% to 0.635% by dataset. We report the percentage of examples screened out by dataset and provide examples of excluded and validated datapoints for each dataset in Appendix A. Many of the excluded instances are clearly mislabeled; the remainder generally represent challenging cases.

## 2.2 CONTEXT WINDOW CONSTRUCTION

We determine the number of examples in the context window by using an estimate of the number of tokens required for each example, for the general task instructions, and for the specific question. Following Yen et al. (2024), we compute these averages using the Llama 2 tokenizer, and use approximately 500K tokens of context for each estimate. We then estimate the number of examples to use by determining how many average-length examples would reach 95% of the target context length.

We sample a distribution over the label classes so that the model cannot use any information about the true distribution over labels (e.g., by guessing that most sentiment classification datasets are balanced). Then, we sample examples to fill this distribution; if there are less total examples than required, we sample with replacement. For each instance, we sample a date and user ID. User IDs are drawn such that 80% of instances have an ID in the 20% of IDs that are most common. Dates are drawn uniformly with replacement from an approximately 40-month range.

Once a context window is constructed, we use it for 25 questions; reusing the same context window allows for prompt caching, reducing the time to evaluate local models and the cost to evaluate on most APIs. At each context length, we sample two context windows per dataset, for a total of 50 questions per dataset or 400 total questions in the test split per context length. We construct OOLONG-synth questions for every power of 2 from 1K to 4M.

**Questions**   We construct three types of OOLONG-synth questions, in order of increasing complexity. *Counting* questions concern simple statistical properties of the label distribution, ranging from identifying the most frequent label to determining the number of examples with each label class. If the label distribution was provided, all tasks in this set should be trivial; these tasks can be seen implicitly as the task of labeling every example and reporting summary statistics. *User information* questions require additional cross-reference with the *user ID* field. *Timeline* questions ask about changes in distribution before or after a certain date, between years, or between months across years. This is more challenging than user information questions because it requires reasoning about *before* or *after* a date, rather than matching a list of IDs (see §B.1 in Appendix for questions).

**Instructions**   For each dataset, we provide a brief (one-sentence) description of the task and label space, as well as the number of examples in-context. We provide instructions at the start and end of the input, as suggested by OpenAI's long context prompting guide.[1] However, we provide the question only at the end, to enable prompt caching.

## 2.3 EVALUATION

**Baseline**   A random baseline is non-trivial for these tasks, because the output space for each question varies. We construct an algorithm for a random baseline with the following rules: (1) in cases where there is an $n$-way choice (e.g. choosing between $n$ labels or choosing a month where some criteria occurs), we select an answer from the set of valid answers at random; (2) if the question requires a numerical answer, we return $N/|L|$, where $N$ is the number of data points in context and $L$ is the labelset; (3) if the question requires a date or user ID, we sample this from the list of dates/IDs in context at random. We compute the random baseline performance as the expected value of this procedure over the dataset.

**Parsing answers**   The task prompts specify an output format. We make a best-effort attempt to parse answers using this format; if this template is not present in the output, we take the candidate answer to be the last substring in the output that appears to match the desired answer type (e.g. a comparison or a number). Generally, this occurs if the model runs out of output token budget before providing an answer.

**Scoring**   For questions that require a label, date, user ID, or comparison (e.g. "greater than" or "less than"), we score on exact match. For questions that require a numerical answer, we set the score to be

$$\mathtt{score}(\hat{y}) = 0.75^{|y-\hat{y}|}$$

This allows for partial credit for answers close to the exact value.

**On iterative solutions**   Although OOLONG-synth allows for a high degree of control over the types of input, these inputs are composed of independent examples from the source dataset. An iterative setup could complete these tasks by calling a model to label each individual example and then deterministically aggregating the resulting labels. We note that this is true for many capability evaluation tasks; for instance, Needle-in-a-Haystack tasks could be solved almost trivially by asking a model if each of the $N$ input sentences individually contains the needle.

Aside from its merits as a capability evaluation, this setting reflects the way many real-world users interact with language models. Real user queries tend to be underspecified and rarely use prompt engineering strategies (Sarkar et al., 2025; Xue et al., 2025). When users do edit prompts, they are more likely to add additional context or more detailed instructions, rather than removing or subdividing tasks (Desmond & Brachman, 2024). This is further exacerbated by recent systems supporting uploads of arbitrary size files for analysis (Google, 2025; Anthropic, 2025). We argue posing questions over a large block of context represents a realistic use scenario. To further motivate the information aggregation problem, we turn to a setting where the input cannot be so trivially decomposed.

---

[1] `https://cookbook.openai.com/examples/gpt4-1_prompting_guide#prompt-organization`

| *Counting* |
| --- |
| Total number of rolls in this episode? |
| What is the count of Crits? (natural rolls of value 1 or 20)? |
| How many {spell type} spells were cast during this episode? |
| How many characters cast {spell name} spell all across episodes? |
| What is the second spell cast in the episode {episode index}? |

| *Enumeration* |
| --- |
| What are the first {count} spells cast in this episode? Return a comma separated list. |
| List the last spell cast in each episode? Return a comma separated list. |

| *Indexing* |
| --- |
| What is the cumulative total of rolls by the end of episode {episode index}? Count the number of rolls and not the values of the rolls. |
| What is the second spell cast in the episode {episode index}? |
| List the last spell cast in each episode? Return a comma separated list. |

Table 2: Question types covered in OOLONG-real dataset.

## 3  OOLONG-REAL

We complement OOLONG-synth with questions derived from real conversational data. OOLONG-real is compiled from the transcripts of a Dungeons and Dragons (D&D) role-playing game, where a group of players collaboratively build a story through in-character actions and success depends on rolling dice. Stories unfold over narratives ("campaigns") that span dozens to hundreds of episodes, with each episode generally lasting 4-5 hours of play. These transcripts involve several levels of conversation, from out-of-character chitchat to rules discussion to in-character actions and speech. Though lightly edited for readability, they reflect naturalistic speech instead of carefully planned written text. Because the conversation is unscripted and can involve tangents or side channels, conversational turns require variable amounts of prior context to resolve. In some instances, the same event (e.g. casting a spell) is discussed for many turns, or brought up again after a long interlude; in other cases, a prior event is "retconned" or revised post hoc.

In Dungeons and Dragons, there are limitations to how frequently characters can take certain actions (e.g., cast certain types of spells). Additionally, fans are often interested in metadata of the play (e.g., whether a certain person is particularly unlucky with their dice rolls or if a character uses a signature spell more or less as the campaign progresses). Because these shows are extremely long and wide-ranging and improvised live, the creators do not plan for or provide this type of information. In lieu of an official source, this information is often annotated meticulously, and with multiple levels of verification, by dedicated fans of the work.

We take this as an example of an information aggregation task in the wild that is *not* simple to reframe as an iterative task. We consider the series Critical Role and the data compiled by CritRoleStats,[2] a fan project that tracked per-episode statistics for the first several campaigns of Critical Role. We devise questions related to characters, dice rolls, and spells cast during episodes and use human-labeled CritRoleStats to compute gold answers.

### 3.1  DATASET COMPILATION

For OOLONG-real, we consider two campaigns from the Critical Role TV series. We used episode transcripts from the Critical Role Dungeons and Dragons Dataset (CRD3) (Rameshkumar & Bailey, 2020), which includes full episode transcripts from the first two campaigns of the Critical Role TV series. For our testbed, we used the first campaign, which consists of 115 episodes. Each line in the transcript includes an utterance with the player name labeled.

---

[2]https://www.critrolestats.com/

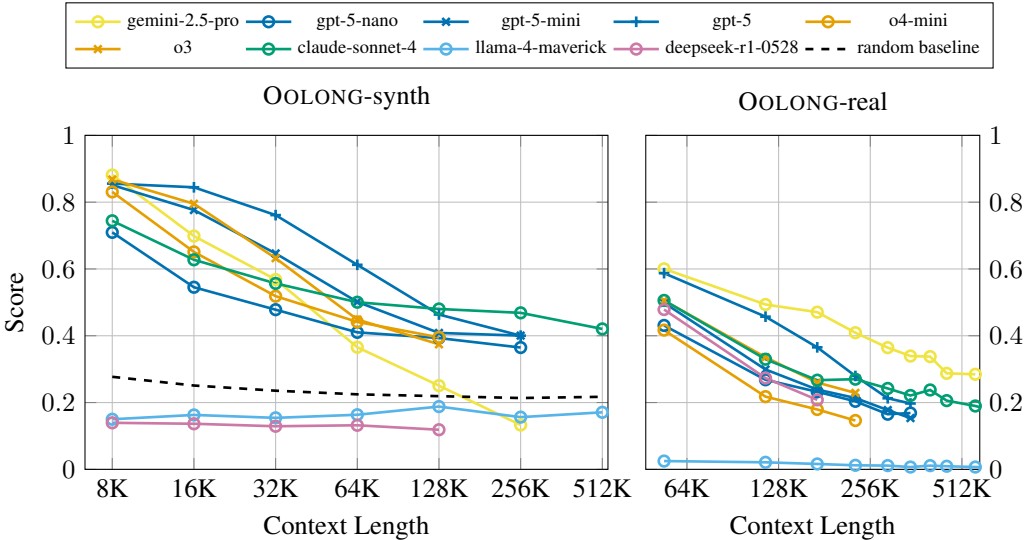

Figure 2: Scores by context window length for OOLONG-synth and OOLONG-real.

**QA pairs**    We use the game statistics compiled by the authors of CritRoleStats. Specifically, we utilize statistics about dice rolls and spells cast in each episode. We design a set of questions that cover a variety of information aggregation tasks; see Table 2 for some examples. We include questions that require processing of single- or multiple-episode transcripts. For multi-episode questions, we concatenate transcripts and use delimiters to highlight the start and end of each transcript. See Table 19 and Table 20 of the Appendix for a full list of OOLONG-real questions. We include varying context windows to evaluate the model's ability to use long-context reasoning and aggregation capabilities. Using a single episode transcript as a context unit, we include context windows ranging from one to 24 episode transcripts. This covers input lengths of 55K to 1.3M tokens.

### 3.2 EVALUATION

We closely follow the evaluation setup from OOLONG-synth.

**Parsing**    Our task prompt requires the model to place the answer in \boxed{}. If the answer cannot be extracted successfully, we attempt to extract the answer for a given question using GPT-5-nano.

**Scoring**    OOLONG-real contains three types of answers: numeric, string, and a list of strings. For numerical answers, we use the same scoring scheme as OOLONG-synth to allow partial credit. We use exact match for string answer types and set overlap for answers of a type list.

## 4    RESULTS AND ANALYSIS

On OOLONG, we benchmark a strong suite of frontier models to study their information aggregation capabilities over long context. We include a mix of models of varying sizes, levels of reasoning, and long-context capabilities.

In Figure 2, we visualize the model performance for context windows up to 512K tokens. As expected, we see a significant drop in performance at higher context windows. The two splits are of similar difficulty at the same context length, although direct comparison is challenging because real data do not necessarily align with fixed context window buckets.[3] A breakdown of results with an aggregate rating over the two datasets is available in the Appendix (Table 4).

---

[3]While performance on OOLONG-real starts lower, this is because the shortest inputs in this dataset are a single episode, with an average length of 55K tokens; models perform similarly on OOLONG-synth around the same context length.

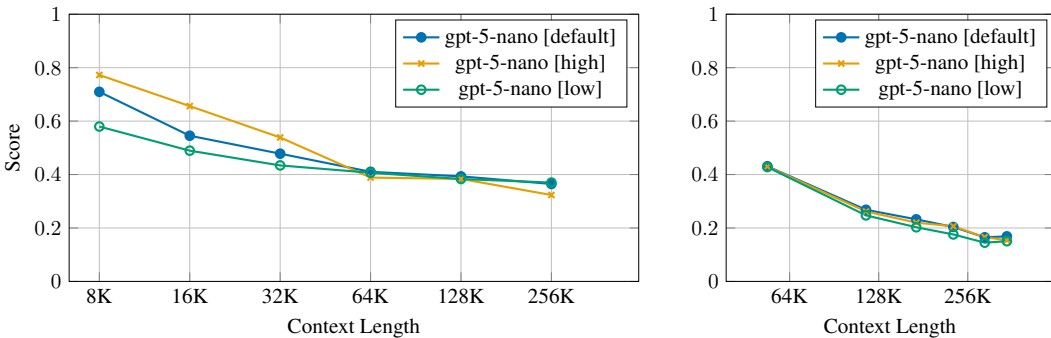

Figure 3: Comparison across reasoning levels.

### 4.1 IMPACT OF REASONING LEVEL

We compare "high" and "low" reasoning effort for GPT-5-nano. Figure 3 shows the results for both the splits of OOLONG. Although OOLONG is a reasoning-intensive task, specifying a higher reasoning effort is only useful for short contexts; after 64k, there is little discernible difference between reasoning levels, with the "high" reasoning setting even slightly underperforming "low" reasoning at 256k on OOLONG-synth. Although it is difficult to draw firm conclusions without the ability to view the reasoning trace, we hypothesize that at context lengths where there is sufficient remaining room in the context window to enumerate labels for each example in-context, adding more reasoning effort may encourage the model to take this strategy. However, since the default routing for this model performs reasonably well at all lengths and especially for longer inputs, we do not explicitly specify a reasoning level for the remainder of the runs.

### 4.2 WHAT TYPES OF QUESTIONS ARE CHALLENGING?

On OOLONG-synth, we further break down results by question type (counting, user, or temporal) and answer type (the expected format of the answer, e.g. as a label or number or date) in Figure 7. Consistently, temporal questions are the most challenging for models, highlighting the difficulty of temporal reasoning. Figure 7 shows that temporal questions are challenging for models; this is reflected in both the performance by task type and the distribution of score by answer type. Questions that require a date or month/year (e.g. "January 2021") as an answer show generally lower performance for the same model, and show greater spread in model capabilities than the other answer categories. For instance, the gap between GPT-5 and GPT-5-nano performance is more than 4x larger for questions that require outputting a date than for questions that require outputting a label. Relative model performance is mostly stable across answer types, although Claude-Sonnet-4 is relatively much stronger on numerical reasoning and comparisons than the GPT series models.

On OOLONG-real, we divide the questions into ten categories: counting, enumeration, frequency, indexing, player, character, roll type/value, spell level/type (Table 19, Table 20). Across models, we find 'frequency' to be the hardest and 'spell type' the easiest (Figure 8). Except for Gemini-2.5-Pro and GPT-5, all models struggle with enumeration and indexing questions. Gemini-2.5 Pro and GPT-5 also make effective use of the player-character mapping provided in the system prompt.

### 4.3 DIFFERENCES BETWEEN OOLONG-SYNTH AND OOLONG-REAL

For most models, the performance trends are consistent across the two datasets. We examine two notable exceptions in more detail.

**Gemini 2.5 Pro** Gemini 2.5 Pro is the strongest model on OOLONG-real, but exhibits performance dropoff at longer contexts on OOLONG-synth. We observe that the model increasingly exceeds its maximum output length during reasoning for long OOLONG-synth inputs. Unlike the other APIs used in benchmarking, the Gemini API does not return any tokens if the max token count is exceeded during reasoning; this results in an automatic score of 0 for any cases of overlong reasoning, whereas

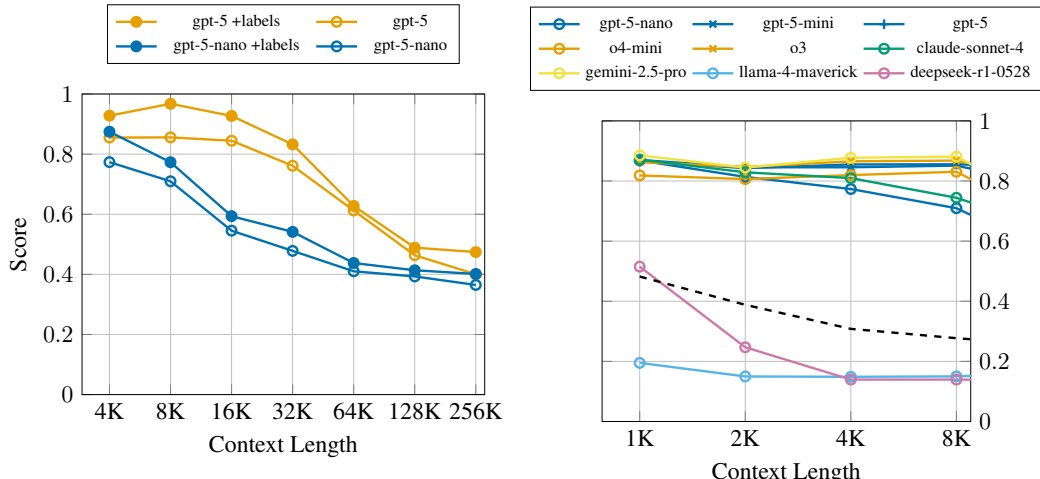

Figure 4: Comparison on OOLONG-synth: (a) we provide the gold labels in the input. This leads to a consistent but small improvement, (b) short context performance; while the top models have similar short-context performance, differences emerge as the context length grows.

a truncated output returned from another model might still score partial credit. Gemini also will return an empty output if the "recitation" filter (indicating substantial regurgitation of pretraining data) is triggered. Because copies of some parts of these inputs (e.g. the IMDB movie reviews) are prevalent on the web, this filter is occasionally triggered when running OOLONG-synth, particularly at longer contexts. We do not observe either of these behaviors on OOLONG-real, and as a result Gemini maintains its strong performance at longer context lengths on this split.

**Deepseek R1** Deepseek R1 is a strong reasoning model and outperforms GPT-5 nano and o4-mini on OOLONG-real. However, on OOLONG-synth, it achieves performance below the random baseline. What causes this discrepancy?

In manual inspection of the traces, we observe pathologies in the reasoning traces for OOLONG-synth that are not present in traces for OOLONG-real and appear only occasionally in traces from other models. We identify these behaviors through discussion between the authors, then prompt GPT-5 nano to label 2,400 randomly selected traces from Deepseek-R1 with a yes/no classification about whether an identified behavior is present in this trace. Many traces (60%) do not provide an answer at all. We hypothesize that, because OOLONG-synth is such an information-dense task, the model's apparent strategy of labeling each example before deciding which are relevant results in running out of context tokens. 64% of traces end in an incomplete sentence.[4] In 17% of cases, the model spends at least some time debating whether the task is impossible or intractable given the length of the input, and in 4% of cases it refuses to respond completely. Appendix C shows example traces for Deepseek-R1 from both OOLONG-synth and OOLONG-real.

### 4.4 SIMPLER SETTINGS FOR OOLONG-SYNTH

**Shorter context** Most models show declining performance with context length. We consider a short-context version of OOLONG-synth, with inputs between 1k and 4k tokens, in Figure 4. While models perform better on this task, several still struggle, showing that even short-context aggregation remains challenging. Performance differences between the top models are difficult to distinguish in the short-context regime, suggesting that these models have the ability to perform the task at some context length. However, no model exceeds 85% performance at any context length.

---

[4]Note that some of these incomplete traces do contain an answer: in a small number of cases, the model provides a candidate answer, then begins to double-check the answer and then exceeds the maximum output length.

**Aggregation without classification** For OOLONG-synth, we can construct an easier version of the task by providing the label for each ICL example in-context. This reduces the task for most questions to simply identifying the relevant instances and summing the occurrence of each label type. Figure 4 shows the behavior of GPT-5 and GPT-5-nano with and without labels provided in-context. As expected, adding labels improves accuracy; however, this improvement ranges from 10.9 points to only 0.79 points. We do not see consistently higher gains from providing labels in longer inputs, which suggests that the lower performance at longer context lengths is not primarily due to an accumulation of mislabeling errors.

Additionally, the improvement is not larger for GPT-5-nano than it is for the more powerful GPT-5, suggesting that the performance difference observed between these models is not due to differing ability to perform the classification task. This is by design; our classification task validation was designed to eliminate overly challenging or misleading examples, as the ability to aggregate information is the main capability we aim to measure.

## 5 RELATED WORK

**Long-context benchmarks** A variety of long-context benchmarks have included some type of aggregation or reasoning-focused task. RULER (Hsieh et al., 2024) benchmark includes synthetic retrieval tasks at varying context lengths. It includes multi-hop tracing and aggregation tasks. HELMET (Yen et al., 2024) expands the tasks in RULER to include tasks related to the downstream use of LMs (reranking, ICL, LongQA, summarization). This includes a set of long-context question-answering tasks from Zhang et al. (2024). LongMemEval (Wu et al., 2025) and multi-round coreference resolution tasks (MRCR) evaluate long-context capabilities with conversational data. MRCR was first introduced in Gemini (Vodrahalli et al., 2024) and was further extended in OpenAI's MRCR (OpenAI, 2025). Unlike the standard NIAH task, the needles and the distractors in MRCR are sampled from the same distributions. OpenAI's MRCR includes variants of 2, 4, and 8 needles, and the task involves retrieving the $i$th instance of one of the needles. The documents in MRCR are synthetic conversations generated using GPT-4o. BABILong (Kuratov et al., 2024) tests long-context reasoning by placing BABI (Weston et al., 2015) reasoning tasks within a long context and evaluating performance as the amount of distractor text increases; however, this cannot vary the amount of information needed to perform the reasoning task, only the amount of distraction present, and thus could be solved by a filter-then-reason approach. TLDM (Hamilton et al., 2025) includes two related tasks about character location and time passed in a narrative setting, but only considers public-domain novels, which likely suffer from data contamination from online discussion of the texts Palavalli et al. (2024). ZeroSCROLLS (Shaham et al., 2023) introduces a task of identifying the percentage of reviews on Amazon for a product that are positive, which is the closest conceptual ancestor of our work; OOLONG-synth encompasses this type of task along with other aggregation tasks. GSM-infinite (Zhou et al., 2025) stress tests long-context reasoning through an adaptable framework for varying task difficulty and length; our work is complementary, as their focus is primarily on increasingly difficult mathematical reasoning through synthetically generated problems, while we focus on a more noisy, naturalistic text setting with less challenging mathematical reasoning required. MoNaCo (Wolfson et al., 2025) considers challenging information seeking queries that require retrieval and aggregation; their emphasis is on retrieval of the necessary documents for intermediate reasoning steps in an agentic setting, a setting completely disjoint from OOLONG. Finally, procedural generation benchmarks (Ye et al., 2025) measure a different type of reasoning task by evaluating the ability to plan and produce coherent long outputs. Table 3 summarizes the distinctions between OOLONG and the most similar prior long-context reasoning benchmarks across a range of desiderata.

**Dungeons & Dragons data** Several prior works consider D&D as a potential testbed for model capabilities; for instance, we use a version of the Critical Role transcripts preprocessed by Rameshkumar & Bailey (2020), who proposed an episode summarization task using fan-written summaries. D&D data has also been used to study theory of mind (Zhou et al., 2023), user assistant development Zhu et al. (2023b), and dialogue generation, with game state either inferred from forum-based games (Callison-Burch et al., 2022) or captured through an external tool (Zhu et al., 2023a); to the best of our knowledge, we are the first to use fan annotations of gold labels and to consider the generation of these statistics as a task in its own right.

| Dataset | Real vs synthetic data | | | Aggregation capabilities | | Measurability | |
|---|---|---|---|---|---|---|---|
| | Realistic inputs | Realistic questions | Relevant text matches distractors | Multi-step reasoning | Numeracy | Must use full input | Easy to eval |
| MRCR | ✓ | ✗ | ✓ | ✓ | ✓ | ✗ | ✓ |
| BABILong | ✗ | ✓ | ✗ | ✓ | ✗ | ✗ | ✓ |
| RULER | ✗ | ✗ | ✗ | † | † | † | ✓ |
| HELMET Summ | ✓ | ✓ | ✓ | ✓ | ✗ | ✓ | ✗ |
| HELMET LongQA | ✓ | ✓ | ✓ | ✓ | ✗ | † | ✓ |
| Oolong-synth | ✗ | ✓ | ✓ | ✓ | ✓ | ✓ | ✓ |
| Oolong-real | ✓ | ✓ | ✓ | ✓ | ✓ | † | ✓ |

Legend: ✓ = yes, ✗ = no, † = in some cases

Table 3: Comparing long-context reasoning benchmarks. OOLONG measures aggregation capabilities in controllable, realistic, and simple-to-evaluate settings.

**Aggregation as an NLP concept** Goldman et al. (2024) argue for the definition of long context tasks in terms of the information dispersion and input scope required to find the answer; under this taxonomy, OOLONG is high dispersion (because relevant info is distribution over the full context length) and high scope (because most of the input is necessary for the task, particularly for OOLONG-synth). DeYoung et al. (2024) measure synthesis in multi-document summarization by asking if summaries convey the consensus opinion of the input documents; this is a related task to the counting tasks in OOLONG-synth, but framed as a measure of summarization capabilities instead of long context capabilities, and thus requires different affordances related to evaluating generated outputs.

# 6 CONCLUSION

We introduce OOLONG, a challenging long-context information aggregation benchmark in two parts. OOLONG-synth uses synthetic aggregation tasks over ICL data to enable finer-grained control of the benchmark settings, while OOLONG-real poses questions over real long-context conversational data and human-annotated labels. On both splits, models struggle, with performance dropping with increasing context length even when controlling for the potential compounding of mislabeling errors.

The low performance on OOLONG suggests more work is necessary in developing both reasoning abilities and more general long context abilities. Even in short context regimes (8K tokens or less), strong models cannot perfectly perform the multi-step reasoning tasks in OOLONG-synth. However, in longer context regimes, many models perform roughly equivalently on OOLONG-synth and OOLONG-real at the same context length, despite the much higher information density in OOLONG-synth. This suggests that better length generalization, regardless of reasoning ability, may also improve performance on this type of challenging long context aggregation task.

However, some models behave differently in extremely information-dense regimes. In particular, both Gemini-2.5-Pro and Deepseek-R1-0528 are strong models that perform well on OOLONG-real but fail in OOLONG-synth because of an over-reasoning phenomenon. This highlights a direction for future work. While prior work on reasoning chains has focused on over-reasoning for adversarial (Kumar et al., 2025) or overly simple problems (Chiang & yi Lee, 2024; Sui et al., 2025), OOLONG indicates that even strong models may struggle to plan the reasoning quantity for information-dense inputs, where it may be desirable to accept a less optimal or more error-prone reasoning strategy (e.g. double-checking less frequently) in order to avoid running into the maximum reasoning tokens. Future models that are aware of the maximum reasoning token budget may be able to more carefully plan reasoning strategies according to the allowed budget.

We see substantial headroom between strong open weights models and API-based models on this task. Overall, our results suggest that there is still a long way to go in designing robust long-context aggregation capabilities for LLMs.

ETHICS STATEMENT

This work is intended to further the evaluation of long-context LLMs. We do not foresee any particular additional risks introduced by our evaluation, although information aggregation, like nearly any LM capability, could be useful for both beneficial and harmful actors.

CritRoleStats, whose data we use as part of constructing gold labels for the OOLONG-real split, explicitly allows the use of their data for statistical analyses in their FAQ, so long as they are credited.[5] We have also notified them of our use of their data.

In the course of this work, we used LM assistance in the writing of some data preprocessing, analysis, and visualization scripts.

REPRODUCIBILITY STATEMENT

We release code for dataset construction and evaluation, in addition to the final dataset version. API-based models typically exhibit some small variability across API calls, and versions of each model may be phased out over time by the providers, so exact replication of outputs may be infeasible; however, we will release the model outputs for each model run reported in the paper.

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

| Model | OOLONG-synth | | | | | | OOLONG-real | | | |
|---|---|---|---|---|---|---|---|---|---|---|
| | Avg. | 8K | 16K | 32K | 64K | 128K | Avg. | 55K | 118K | 175K |
| GPT-5 | **70.75** | 85.56 | **84.45** | **76.12** | **61.24** | 46.36 | 47.00 | 58.74 | 45.72 | 36.53 |
| Gemini-2.5-Pro | 55.29 | **88.13** | 69.84 | 56.83 | 36.56 | 25.06 | **52.16** | **60.04** | **49.36** | **47.07** |
| o3 | 62.37 | 86.80 | 79.52 | 63.23 | 44.86 | 37.45 | 36.71 | 50.57 | 33.57 | 25.99 |
| GPT-5-mini | 63.68 | 85.13 | 77.65 | 64.64 | 50.14 | 40.85 | 34.55 | 49.86 | 29.90 | 23.89 |
| Claude-Sonnet-4 | 58.18 | 74.43 | 62.75 | 55.67 | 50.04 | **48.02** | 36.75 | 50.58 | 32.98 | 26.70 |
| o4-mini | 56.74 | 83.07 | 65.10 | 51.86 | 44.15 | 39.53 | 27.13 | 41.69 | 21.77 | 17.93 |
| GPT-5-nano | 50.73 | 70.96 | 54.53 | 47.81 | 41.02 | 39.31 | 31.05 | 43.09 | 26.82 | 23.23 |
| Deepseek-R1 | 13.11 | 13.94 | 13.65 | 12.91 | 13.20 | 11.87 | 32.00 | 47.85 | 27.35 | 20.81 |
| Llama-4-Maverick | 16.37 | 15.00 | 16.29 | 15.42 | 16.35 | 18.80 | 2.07 | 2.48 | 2.11 | 1.62 |

Table 4: OOLONG results on a number of strong models. All models we test support at least 200K context; thus, we report OOLONG scores as an average over scores on 8K-175K inputs. OO-LONG-synth is a more information-dense task than OOLONG-real; the ranking of some models shifts slightly between the two settings. Models are sorted by the average between the two benchmarks.

| Dataset | % Removed |
|---|---|
| Spam | 0.635% |
| TREC-Q-coarse | 0.048% |
| AGNews | 0.026% |
| App Reviews | 0.051% |
| Formality | 0.108% |
| IMDB | 0.042% |
| Metaphors | 0.000% |
| MultiNLI | 0.109% |
| Negation | 0.016% |
| Yahoo Topics | 0.188% |

Table 5: Only a small fraction of examples are removed during validation for each source dataset.

## A  ICL LABEL VALIDATION

Table 1 shows the fraction of examples removed by the validation step for each dataset.

For each ICL dataset used to construct OOLONG-synth, we provide example instances that failed validation (and thus were discarded) and that passed validation and were used for constructing context windows. These examples are selected nearly-randomly; many of the instances that fail validation contain sexual content or offensive language, and we screen these out of the examples shown wherever possible. While the instances screened out vary by dataset, we note that the datasets that use labels scraped from web content (i.e. Yahoo Topics, AGNews, IMDB reviews, App Reviews) generally have higher rates of validation failures. Some of these failures appear to be genuine mis-labelings, which is likely because of noise in the user behavior (e.g. a user asks a question about Business & Finance but mistakenly posts it in the Sports topic, or a user writes a review with mostly negative text but gives the product 4 stars).

| Input | Label | Passed validation? |
|---|---|---|
| 'Virtual Girlfriend' Demands Gifts HONG KONG - She needs to be coddled with sweet talk and pampered with gifts, but you'll never see her in the flesh. A Hong Kong company has developed a "virtual girlfriend" for new cell phones with video capability... | World | ✗ |
| Court Hears Case of Brain Damaged Woman (AP) AP - The Florida Supreme Court questioned lawyers Tuesday about the extent of the power handed to Gov. Jeb Bush under a law that let him order the reinsertion of a brain-damaged woman's feeding tube. | Sci/Tech | ✗ |
| Rooney backs fiercesome threesome (AFP) AFP - Wayne Rooney believes the three-pronged attack of himself, Michael Owen and Jermain Defoe can put England on the fast track to the World Cup finals. | World | ✗ |
| From mouths of babes: What's hot, what's not How hot are the "Hot Dozen" toys? To find out, The Boston Globe put six toys from the 2004 Toy Wishes magazine Hot Dozen list in front of 13 kids from the Charlestown Boys and Girls Club. | Business | ✗ |
| Crematory operator to get 12 years Ray Brent Marsh, who is to enter the plea Friday, had faced up to 8,000 years in a case that shocked the nation two years ago when investigators found hundreds of corpses at his rural northwest facility. | Business | ✗ |
| Vending Machines Making Room for Healthy Products WASHINGTON (AP) – The typical vending machine fare consists of chocolate bars and potato chips, leaving few options for people seeking low-calorie or low-salt snacks. That is changing now as companies develop markets for products they expect to satisfy both nutritionists and consumers... | Sci/Tech | ✓ |
| Venezuelans Line Up to Vote on Chavez CARACAS, Venezuela - Summoned by bugle calls and the detonations of huge firecrackers, Venezuelans turned out in unprecedented numbers Sunday to vote on whether to force leftist President Hugo Chavez from office. Some lines at polling places extended for 1.25 miles... | World | ✓ |
| Singh Snares PGA Title Vijay Singh outlasts Justin Leonard and Chris DiMarco in a three-way playoff to win the PGA Championship on Sunday at Whistling Straits in Haven, Wisconsin. | Sports | ✓ |
| China's Panchen Lama visits Tibet The boy named by the Chinese authorities as the 11th Panchen Lama visits a temple in Tibet. | World | ✓ |
| The Region's Highest-Paid Executives Pay for the Washington area's top executives rose significantly last year, reversing the downward trend that set in with the recession in 2001. | Business | ✓ |

Table 6: AGNews examples that failed and passed validation.

| Input | Label | Passed validation? |
|---|---|---|
| Kill Me.. | positive | ✗ |
| Used to be great app. But since the last 7 updates always get - Authentication failed for: x@gmail.com | positive | ✗ |
| App force closing on changing THEME PLZ FIX! | positive | ✗ |
| You can't uninstall & I don't like my space being used. But am glad for the blind & deaf | positive | ✗ |
| GOOGLE TALKBACK Read carefully & then give at least a good reviews this app is made only for persons who got disability such as a blind person & if you want to get rid this simply just ROOT your device dumb head!! | negative | ✗ |
| This thing is great. I am at the bottom of the learning curve but in the few minutes I have played with it it seems like it will be easy to learn and to use. Just what I hoped for. | positive | ✓ |
| Crash when set a demodulation! :( | negative | ✓ |
| doesn't work with rtl-2832 Says it works with rtl devices but doesn't. never displays anything even though the rtl 2832 works with other spectrum analyzers. no help files anywhere so good luck figuring out what's wrong! | negative | ✓ |
| Great app! Works really well with Galaxy note 5 | positive | ✓ |
| Works great I am using HackRF One with this apps. | positive | ✓ |

Table 7: App Reviews examples that failed and passed validation.

## B  QUESTION TYPES

### B.1  OOLONG-SYNTH

Table 16, Table 17, and Table 18 list the questions used for each type of task in OOLONG-synth.

### B.2  OOLONG-REAL

Table 19 and Table 20 lists the question types used for single and multi-episode settings in OOLONG-real dataset.

| Input | Label | Passed validation? |
|---|---|---|
| The Unit secret agent has signed on to appear in three episodes as a businessman being shown some houses - and, presumably, one bedroom in particular - by our randy realtor heroine. | formal | ✗ |
| Barely 12, with large brown eyes and stick-like arms, Fandi is 3 years older than his brother – in his eyes almost a man. | formal | ✗ |
| D-Lister Avril Lavigne appears on the cover of Z-List magazine Savvy this month. | formal | ✗ |
| ORHS is going through it now, but from all I can perceive the district is out for the quick fix, chop off the principal, instead of really trying to assist the students and dig to the bottom to find the truth. | formal | ✗ |
| Little Sally Draper (Kiernan Shipka) is a "Patty Duke"-era girl - indeed, "Mad Men" is currently set during the month "The Patty Duke Show" premiered in 1963 - and on a "Patty Duke"-like show, she'd just be a lispy Shirley Temple doll with a crush on Daddy. | formal | ✗ |
| I have tried everything possible to attract business. | formal | ✓ |
| Never heard that one before. | informal | ✓ |
| "Yup." | informal | ✓ |
| That's how we pass our traits to the next generation: through DNA | formal | ✓ |
| Wedding rituals differ in different regions and communities in India. | formal | ✓ |

Table 8: Formality examples that failed and passed validation.

| Input | Label | Passed validation? |
|---|---|---|
| Good movie, very 70s, you can not expect much from a film like this. Sirpa Lane is an actress of erotic films, a nice body but nothing exceptional. Not demand a lot from these films are light years away from the movies today, the world has changed incredibly. The plot is simple and the actors not extraordinary. | positive | ✗ |
| This film Evil Breed: The legend of samhain contains very little thought or effort. It is ridiculed with specs of ultra fast "slasher" style death and plain disgusting acts of death. The acting was rated a D as the actors show very little ability, and the stupidity of them in the film is too questionable. | positive | ✗ |
| The movie "Holly" may make the audience want to donate money towards organizations that improve the life for these poor youngsters, but the film's dramatic weaknesses may reduce its chances of being seen by enough people to make a difference. Overall, I think the concept is better as a documentary and it was not as touching as a movie. | positive | ✗ |
| Although this film put Davis on the map due to her brilliantly intense performance, this film is strangely unsatisfying to me as a whole. What I cannot fathom for the life of me is just how or why Phillip would take the constant abuse this tramp constantly dishes out towards him. | positive | ✗ |
| This film is just plain horrible. John Ritter doing pratt falls, 75% of the actors delivering their lines as if they were reading them from cue cards, poor editing, horrible sound mixing, and a plot that really goes nowhere. If I could sum this film up in one word, that word would be: Suckotrocity | negative | ✓ |
| Zentropa has much in common with The Third Man, another noir-like film set among the rubble of postwar Europe. Like TTM, there is much inventive camera work. There is an innocent American who gets emotionally involved with a woman he doesn't really understand, and whose naivety is all the more striking in contrast with the natives. But I'd have to say that The Third Man has a more well-crafted storyline. Zentropa is a bit disjointed in this respect. Perhaps this is intentional: it is presented as a dream/nightmare, and making it too coherent would spoil the effect. This movie is unrelentingly grim–"noir" in more than one sense; one never sees the sun shine. Grim, but intriguing, and frightening. | positive | ✓ |
| Never in my life have I come across a movie as bad as The Zombie Chronicles. Filmed on a budget of what looks to be about 20 bucks, TZC is a completely horrible horror movie that relies on lame, forgettable actors. Simply put, avoid TZC like a sexually-transmitted disease. | negative | ✓ |
| Without wishing to be a killjoy, Brad Sykes is responsible for at least two of the most dull and clichéd films I've ever seen. The acting is terrible, the print is shoddy, and everything about this film screams "seriously, you could do better yourself". | negative | ✓ |

Table 9: IMDB examples that failed and passed validation. Only eight examples are shown because of the example length.

| Input | Label | Passed validation? |
|---|---|---|
| Draupadi's eyes were diamonds. ↔ Draupadi's eyes were beautiful. | correct | ✓ |
| The faculty meeting was an easy breeze ↔ The faculty meeting was very easy and relaxing | correct | ✓ |
| The house was a tomb. ↔ The house was big. | incorrect | ✓ |
| I had already planted the idea in her mind. ↔ I had already scared her about the idea. | incorrect | ✓ |
| It is sad to observe the fruits of ignorance. ↔ It is sad to observe the effects of ignorance. | correct | ✓ |

Table 10: Metaphors examples that passed validation. No examples failed validation for this dataset.

| Input | Label | Passed validation? |
|---|---|---|
| You're safe. → You have nothing to worry about. | neutral | ✗ |
| asks Burton, cupping his ear. → Burton wanted to ask what was going on, but he couldn't do much of anything while bound and gagged. | contradiction | ✗ |
| I "poison my dearest Emily!" → How could you think I would poison Emily? | entailment | ✗ |
| When he finally succeeded, after a prolonged siege and heavy losses, he punished the local population by cutting off the noses and lips of all men except those who played wind instruments. → All men who played wind instruments were tasked with helping cut off people's noses. | neutral | ✗ |
| They died slowly, their eyes bulging and faces turning blue. → They died asphyxiated. | neutral | ✗ |
| well you see that on television also → You can see that on television, as well. | entailment | ✓ |
| Vrenna and I both fought him and he nearly took us. → Neither Vrenna nor myself have ever fought him. | contradiction | ✓ |
| This analysis pooled estimates from these two studies to develop a C-R function linking PM to chronic bronchitis. → The analysis proves that there is no link between PM and bronchitis. | contradiction | ✓ |
| He turned and smiled at Vrenna. → He smiled at Vrenna who was walking slowly behind him with her mother. | neutral | ✓ |
| We sought to identify practices that were commonly implemented by the agencies within the past 5 years. → We want to identify practices commonly used by agencies in the last 5 years | entailment | ✓ |

Table 11: MultiNLI examples that failed and passed validation.

| Input | Label | Passed validation? |
|---|---|---|
| No care is anything that serves as an enticement. | True | ✗ |
| No care may be anything that serves as an enticement. | True | ✗ |
| A loud utterance is an appropriate definition of produce in no context. | True | ✗ |
| Complete attention is an appropriate definition of candidate in no context. | True | ✗ |
| No stress is a message received and understood. | True | ✗ |
| Action refers to a military engagement. | True | ✓ |
| Action never stands for a distinguishing quality. | True | ✓ |
| Action does not stand for a military engagement. | False | ✓ |
| Action may stand for a military engagement. | True | ✓ |
| Not a single action is a military engagement. | False | ✓ |

Table 12: Negation examples that failed and passed validation.

| Input | Label | Passed validation? |
|---|---|---|
| Go chase after her and run her over while she's crossing the street | ham | ✗ |
| i want to grasp your pretty booty :) | ham | ✗ |
| No da if you run that it activate the full version da. | ham | ✗ |
| i am seeking a lady in the street and a freak in the sheets. Is that you? | ham | ✗ |
| Should i send you naughty pix? :) | ham | ✗ |
| WINNER!! As a valued network customer you have been selected to receive a £900 prize reward! To claim call 09061701461. Claim code KL341. Valid 12 hours only. | spam | ✓ |
| Had your mobile 11 months or more? U R entitled to Update to the latest colour mobiles with camera for Free! Call The Mobile Update Co FREE on 08002986030 | spam | ✓ |
| I'm gonna be home soon and i don't want to talk about this stuff anymore tonight, k? I've cried enough today. | ham | ✓ |
| SIX chances to win CASH! From 100 to 20,000 pounds txt¿ CSH11 and send to 87575. Cost 150p/day, 6days, 16+ TsandCs apply Reply HL 4 info | spam | ✓ |
| I HAVE A DATE ON SUNDAY WITH WILL!! | ham | ✓ |

Table 13: Spam examples that failed and passed validation. The examples that failed validation were almost exclusively ham messages with sexual content; the selected messages are the least inappropriate of the examples that failed validation.

| Input | Label | Passed validation? |
|---|---|---|
| What explosive do you get by mixing charcoal, sulfur and saltpeter? | entity | ✗ |
| What is the procedure called for drilling a hole in your skull to achieve a higher consciousness? | entity | ✗ |
| Name 11 famous martyrs. | human being | ✓ |
| What's the Olympic motto? | description and abstract concept | ✓ |
| What is the highest waterfall in the United States? | location | ✓ |
| What does the abbreviation AIDS stand for? | abbreviation | ✓ |
| How many points make up a perfect fivepin bowling score? | numeric value | ✓ |

Table 14: TREC-coarse-Q examples that failed and passed validation. Only two examples failed validation.

| Input | Label | Passed validation? |
|---|---|---|
| Is there a God? The question to end all questions, and begin them. | Business & Finance | ✗ |
| why do we need to lie? | Health | ✗ |
| try to type the word supercalifragilisticexpialidocious. 20 times fast you can only make 10 mistakes? This is for fun! | Education & Reference | ✗ |
| do you think it is okay to tell a lie? | Education & Reference | ✗ |
| where can you purchase cesium carbonate? | Science & Mathematics | ✗ |
| What are good sources to find out about new gospel artists? Is there a site that focuses primarily on gospel? | Entertainment & Music | ✓ |
| How a black hole is formed? I would like to know how a black hole can possibly be formed. Are there any experimental evidence of such creation? | Science & Mathematics | ✓ |
| Economics of running a restaurant? Running a restaurant looks like hard work and long hours. What percentage of restaurants are profitable? | Business & Finance | ✓ |
| Why doesn't the NBA implement a minor leagues? I don't want to see any more High School kids on the court. | Sports | ✓ |
| how does a helicopter fly this is a miracle- I always wanted to learn to fly one of those. Can someone explain how can it get airborne? | Science & Mathematics | ✓ |

Table 15: Yahoo examples that failed and passed validation. One class of examples that failed validation, not demonstrated here, are sexually explicit questions labeled as seemingly random categories (likely from spam posters on the original Yahoo Answers forum).

| Counting |
| --- |
| In the above data, which of the labels is the most common? Give your final answer in the form 'label: answer' where answer is one of the labels: {label_list}. |
| In the above data, which of the labels is the least common? Give your final answer in the form 'label: answer' where answer is one of the labels: {label_list}. |
| In the above data, is label '{A}' more common, less common, or the same frequency as label '{B}'? Give your final answer in the form 'Answer: {A} is [X]{B}', where [X]is 'more common than', 'less common than', or 'same frequency as'. |
| In the above data, how many data points should be classified as label '{label}'? Give your final answer in the form 'Answer: number'. |

Table 16: Counting questions used in the OOLONG-synth dataset.

| User |
| --- |
| In the above data, which user is represented most often? Give your final answer in the form 'User: [X]', where [X]is the user ID. |
| In the above data, which user is represented the second most often? Give your final answer in the form 'User: [X]', where [X]is the user ID. |
| For the following question, only consider the subset of users with IDs {user_names}. Among these users, which user is represented most often? Give your final answer in the form 'User: [X]', where [X]is the user ID. |
| For the following question, only consider the subset of users with IDs {user_names}. Among these users, which user is represented the second most often? Give your final answer in the form 'User: [X]', where [X]is the user ID. |
| For the following question, only consider the subset of instances that are associated with user IDs {user_names}. Among instances associated with these users, {any of the Counting questions above} |
| For the following question, only consider the subset of users with IDs {user_names}. Among these users, which user has the most instances with the label {label}? Give your final answer in the form 'User: [X]', where [X]is the user ID. |
| In the above data, which user has the most instances with the label {label}? Give your final answer in the form 'User: [X]', where [X]is the user ID. |
| In the above data, which user has more instances with the label {label}: User {A} or User {B}? Give your final answer in the form 'User: [X]', where [X]is the user ID. |

Table 17: User questions used in the OOLONG-synth dataset.

| Timeline |
| --- |
| In the above data, which date is represented most often? Give your final answer in the form 'Date: [X]', where [X]is the date in the format MM/DD/YYYY. |
| In the above data, which date is represented most often? Give your final answer in the form 'Date: [X]', where [X]is the date in the format MM/DD/YYYY. |
| In the above data, which date is represented the second most often? Give your final answer in the form 'Date: [X]', where [X]is the date in the format MM/DD/YYYY. |
| In the above data, how many dates are represented exactly {n} times? Give your final answer in the form 'Answer: [X]', where [X]is the number of dates represented exactly {n} times. |
| In the above data, was label '{key}' more common, less common, or the same frequency before {time}, as compared to after {time}? Give your final answer in the form 'Answer: {key} is [X]before {time}', where [X]is 'more common', 'less common', or 'the same frequency'. |
| In the above data, was label '{key}' more common, less common, or the same frequency before {time}, as compared to after {time}? Give your final answer in the form 'Answer: {key} is [X]before {time}', where [X]is 'more common', 'less common', or 'the same frequency'. |
| For the following question, only consider the subset of instances that occur in {month_name} of any year. Among instances occuring in {month_name},{any of the Counting questions above} |
| For the following question, only consider the subset of instances that occur between {starting_date} and {ending_date}, inclusive. Among instances occuring in this date range, {any of the Counting questions above} |
| In which month did the label '{label1} first occur more often than the label '{label2}'? Give your final answer in the form 'Answer: [month][year]', where [month]is the name of the month and [year]is the four-digit year where '{label1}' first occured more often than '{label2}.' |
| For how many months does the label '{label1}' occur more frequently than the label '{label2}'? Disregard months where there is a tie. Give your final answer in the form 'Answer: [X]', where [X]is the number of months where '{label1}' occurs more often than '{label2}.' |
| For how many months is the label '{label}' the single most frequently occuring label? Disregard months where there is a tie for the most common label. Give your final answer in the form 'Answer: [X]', where [X]is the number of months where '{label}' is the most common label. |

Table 18: Timeline questions used in the OOLONG-synth dataset.

| Rolls |
| --- |
| Total number of rolls in this episode? (counting) |
| Total number of rolls by the character {character name} in this episode? (counting, character) |
| Total number of rolls by the player {player name} in this episode? (counting, player) |
| Total number of rolls of type {roll type} in this episode? (counting, roll type) |
| Number of rolls of natural value {roll value} in this episode? (counting, roll value) |
| In this episode, what percentage of rolls were of value {roll value}? round to the nearest integer. (counting, roll value) |
| What is the most common roll type in this episode? Return a comma separated list. (frequency) |
| What is the least common roll type in this episode? Only include types with more than one roll. Return a comma separated list. (frequency) |
| What is the most common natural roll value in this episode? Return a comma separated list. (frequency) |
| What is the least common natural roll value in this episode? Only include values with more than one roll. Return a comma separated list. (frequency) |
| What is the count of Crits? (natural rolls of value 1 or 20)? (counting, roll value) |
| What is the count of Nat20s (natural rolls of value 20)? (counting, roll value) |
| What is the count of Nat1s (natural rolls of value 1)? (counting, roll value) |

| Spells |
| --- |
| How many spells were cast during this episode? (counting) |
| How many spells were cast by the character {character name} in this episode? (counting, character) |
| How many spells were cast by the player {player name} in this episode? (counting, player) |
| How many {spell type} spells were cast during this episode? (counting, spell type) |
| What is the first spell cast in this episode? (enumeration) |
| What are the first two spells cast in this episode? (enumeration) |
| What are the first three spells cast in this episode? (enumeration) |
| What is the last spell cast in this episode? (enumeration) |
| What are the last two spells cast in this episode? (enumeration) |
| What are the last three spells cast in this episode? (enumeration) |
| What is the first spell cast by each character in this episode? Return a comma separated list and retain the order of spells as they appear in the episode. (enumeration) |
| What is the last spell cast by each character in this episode? Return a comma separated list and retain the order of spells as they appear in the episode. (enumeration) |
| How many characters cast {spell name} spell in this episode? (counting, spell type) |
| What is the most common spell in this episode? Return a comma separated list. (frequency) |
| What is the least common spell in this episode? Only include spells that were cast at least once. Return a comma separated list. (frequency) |
| Which spells were cast by more than one character in this episode? Return a comma separated list. (counting, enumeration) |
| What is the total number of cantrip spells cast in this episode? (counting, spell type) |
| In this episode, how many times was a spell cast at a level higher than its base level? (counting) |
| In this episode, which spells were cast at a level higher than their base level? Return a comma separated list of unique spells. (enumeration) |

Table 19: Question types used in the OOLONG-real dataset (single episode). For reference, we provide the question types in parentheses.

## C  DEEPSEEK R1 TRACE EXAMPLES

Figure 5 and Figure 6 contrast the behavior of this model on representative examples from the OOLONG-synth and OOLONG-real splits. See the main text for more discussion.

| Rolls |
|---|
| What is the cummulative total of rolls by the end of episode {episode index}? Count the number of rolls and not the values of the rolls. (counting) |
| What is the cummulative total of rolls by the character {character name} at the end of episode {episode index}? Count the number of rolls and not the values of the rolls. (counting, character) |
| Total number of rolls across all the episodes? (counting) |
| Total number of rolls by the character {character name} across all episodes? (counting, character) |
| Total number of rolls by the player {player name} across all episodes? (counting, player) |
| Total number of rolls of type {roll type} across all episodes? (counting, roll type) |
| Number of rolls of natural value {roll value} across all episodes? (counting, roll value) |
| Across all episodes, what percentage of rolls were of value {roll value}? round to the nearest integer. (counting, roll value) |
| What is the most common roll type across all episodes? Return a comma separated list. (frequency) |
| What is the least common roll type across all episodes? Only include types with more than one roll. Return a comma separated list. (frequency) |
| What is the most common natural roll value across all episodes? Return a comma separated list. (frequency) |
| What is the least common natural roll value across all episodes? Only include values with more than one roll. Return a comma separated list. (frequency) |
| What is the total count of Crits across all episodes? (natural rolls of value 1 or 20)? (counting, roll value) |
| What is the total count of Nat20s across all episodes? (natural rolls of value 20)? (counting, roll value) |
| What is the total count of Nat1s across all episodes? (natural rolls of value 1)? (counting, roll value) |

| Spells |
|---|
| What is the cummulative total of spells cast by the end of episode {episode index}? (counting, indexing) |
| What is the first spell cast in the episode {episode index}? (enumeration, indexing) |
| What is the second spell cast in the episode {episode index}? (enumeration, indexing) |
| What is the third spell cast in the episode {episode index}? (enumeration, indexing) |
| List the first spell cast in each episode? Return a comma separated list. (enumeration) |
| List the last spell cast in each episode? Return a comma separated list. (enumeration) |
| List the first spell cast by the character {character name} in each episode? Return a comma separated list. (enumeration, character) |
| List the last spell cast by the character {character name} in each episode? Return a comma separated list. (enumeration, character) |
| How many spells were cast across all episodes? (counting) |
| How many spells were cast by the character {character name} across all episodes? (counting, character) |
| How many spells were cast by the player {player name} across all episodes? (counting, player) |
| How many {spell type} spells were cast across all episodes? (counting, spell type) |
| How many characters cast {spell name} spell across all episodes? (counting, spell type) |
| What is the most common spell across all episodes? Return a comma separated list. (frequency, enumeration) |
| What is the least common spell across all episodes? Only include spells that were cast at least once. Return a comma separated list. (frequency, enumeration) |
| What is the total number of cantrip spells cast across all episodes? (counting, spell type) |
| Across all episodes, how many times was a spell cast at a level higher than its base level? (counting, spell level) |
| Across all episodes, which spells were cast at a level higher than their base level? Return a comma separated list of unique spells. (enumeration, spell level) |

Table 20: Question types used in the OOLONG-real dataset (multi-episode). For reference, we provide the question types in parentheses.

<think> We are given 21 pairs. We need to classify each pair as one of 'neutral', 'contradiction', or 'entailment'. Then, we will count the frequency of 'contradiction' and 'entailment' and compare them. Let's go through each pair one by one: 1. "and then the class uh double A Dallas Fort Worth Spurs we later uh they went to the Texas league - Dallas Fort Worth Spurs got into the Texas League." - The first sentence says they went to the Texas league. The second says they got into the Texas League. This seems to be the same meaning. So, entailment. 2. "Specifically, although most public companies are required to have a board that is comprised of a majority of independent directors and certain key committees are required to be comprised solely of outside directors, a closer look reveals that the independence of many boards may be more a matter of form than substance. - Public companies can put whoever they want on their boards." - The first sentence says that public companies are required to have independent directors, but the independence might be in form only. The second sentence says they can put whoever they want. This contradicts the requirement mentioned. So, contradiction. **[....authors truncated classifications for space]** 19. "And the bureaucratic approach prevents the section as a whole from taking positions on–or creating–literary issues. - Bureaucracy makes groups not take positions or a stand on literary issues." - The second is a restatement of the first. Entailment. 20. "They are linked not only by geographical location but also by here are the upmarket and fashionable neighborhoods, with affluent residential sections, trendy restaurants, and L.A.'s fabulous shopping areas. - They are linked because they both have very exclusive shoe stores." - The first says they are linked by upmarket and fashionable neighborhoods, etc. The second says they are linked because of exclusive shoe stores. The first does not mention shoe stores. So, the second is a specific reason that is not stated. This could be neutral. However, note that the first lists shopping areas, which might include shoe stores, but it doesn't specify. The second is too

Figure 5: Example reasoning trace for OOLONG-synth. The model fails to complete the task because it runs out of tokens during the reasoning trace. The majority of the trace (classifying items 3-18) is omitted for space.

Based on the provided transcript of Episode 2, the second spell cast is Disintegrate, cast by Lyra (Felicia Day's character).

Sequence of Events: 1. First Spell: Vex'ahlia (Laura Bailey) casts Hunter's Mark on a frost giant (during her turn in combat). 2. Second Spell: Lyra (Felicia Day) casts Disintegrate on the same frost giant (during her subsequent turn).

Key Excerpt: Felicia: "I have a spell called Disintegrate. Can I cast that on its wing? [...] I'm doing Disintegrate. And it's 10d6 plus a d10, plus 40." Matt: "Disintegrate is nasty. [...] As you release the torrent of electrical energy, it courses through the body [...] the giant looks severely messed up."

Thus, the second spell in Episode 2 is Disintegrate.

Disintegrate

Figure 6: Example reasoning trace for OOLONG-real. The model completes the task within a relatively short reasoning trace, despite this task using substantially longer input.

# D    QUESTION TYPE ANALYSIS

## D.1    OOLONG-SYNTH

In Figure 7, we report the model performance by question and task types on OOLONG-synth.

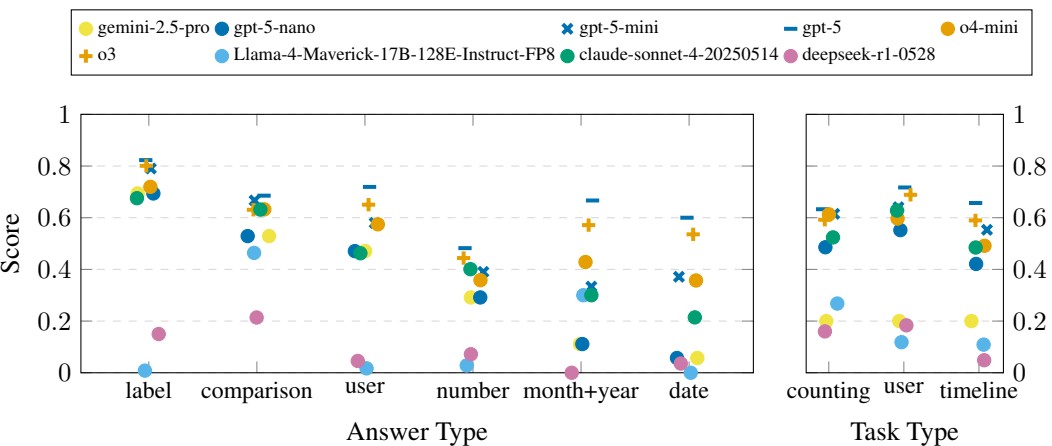

Figure 7: The performance trend for models by type of answer and type of task on OOLONG-synth.

## D.2    OOLONG-REAL

In Figure 8, we report model performance by question type on OOLONG-real.

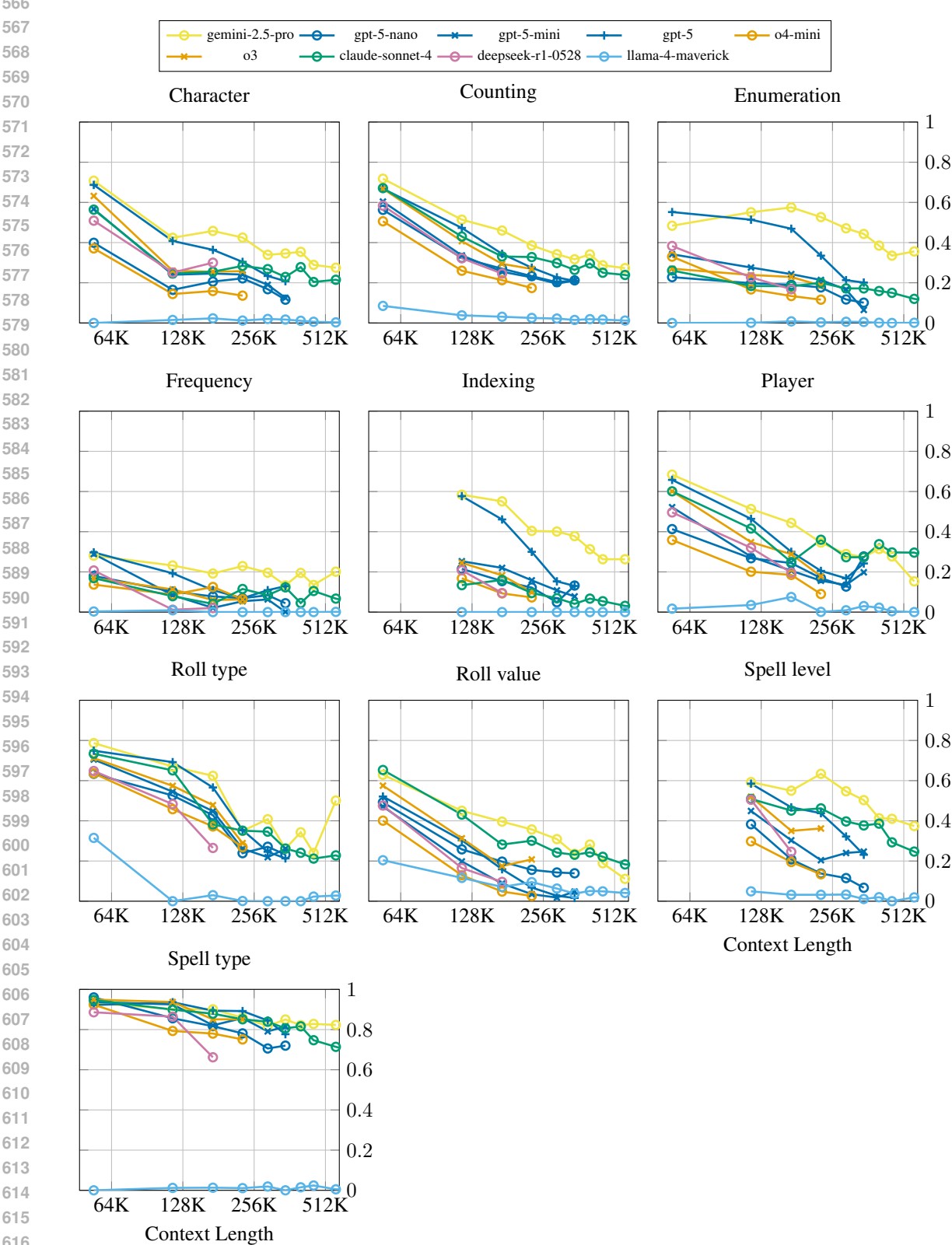

Figure 8: Scores by context window length for various question types in OOLONG-real. Some questions are only relevant for multi-episode settings.

# E OOLONG-REAL DATA

## E.1 DATA QUALITY AND CURATION

The episode transcripts were first hand-transcribed by members of the fan community, who implemented a multi-step editing process with editorial guidelines. These were then further cleaned by Rameshkumar & Bailey (2020) to resolve any remaining inconsistencies. We refer the reader to the original paper for more details on the curation process. The stats were manually compiled by volunteers and the team at CritRoleStats. While it's not possible to completely eliminate sources of potential error, these types of fan-documentation processes are fastidiously annotated and cross-checked by multiple people– a standard of care far higher than that of a paid annotator with no connection to the task or data domain.

## E.2 CHALLENGES OF OOLONG-REAL

In the tables below, we highlight various unique challenges presented by OOLONG-real. Figure 9 covers text that includes in-character, out-of-character and narrative content. Additionally, we show complex discussions surrounding dice rolls (Figure 10, Figure 12) and damage calculation (Figure 11).

---

**Matt:** All right. As your smaller friend curls awake, Sam, would you like to describe your character?
**Sam:** Yeah. Um. I am a little goblin girl. (laughter)
**Sam:** I am a goblin. So, you know, the green skin, the green hair, the yellow eyes. And she wears not-great clothes, just like her traveling companion there. She hides in the shadows a lot, because she knows goblins aren't welcome in this part, and that's about it. I mean, she's a little skittish, and right now she's probably stirring awake as well, right?
**Matt:** Well. What's your name?
**Sam:** Oh. Nott the Brave.
**Matt:** So Caleb, as you come to consciousness, you glance over and can see, slowly snoring and rousing at about the same time, Nott's eyes blink open, her slowly groaning face looking over towards you.
**Sam:** (high-pitched Cockney accent) Oh! You're finally awake, I see. Oh yeah, motherfuckers. It's on. You were out for quite some time, there. Rough day, eh?
**Liam:** (light German accent) Not our best day, no.
**Sam:** No, I mean, usually you're so good at everything, but yesterday you were just– maybe you needed the sleep, is what you needed.

Figure 9: Example of mixed in-character, out-of-character, and narrative text.

---

**Matt**: The old mans̀ body stops quaking, his flesh now grey and mangled like an ancient tree trunk. He turns, his eyes blood-red and bulging, his lips curled into a horrifying grimace. The two Crowns̀ Guard begin to try and make their way through the panicked crowd, but the people, like a wave of chaos, are keeping them at bay. I need everyone to roll initiative. (yelling)

**Liam**: The miniatures come out, guys!

**Travis**: First map! (yelling)

**Marisha**: I rolled a natural one.

**Sam**: That bodes well.

**Matt**: So you guys, this guy is actually over here. You guys are all up here in the front.

**Laura**: Oh yeah, we were right in the fucking front row. Look at us! Wère so cute!

**Taliesin**: Oh my god, itś so pretty.

**Sam**: Wère all going to die.

**Liam**: Probably. But then the third campaign begins.

**Laura**: Yay.

**Sam**: I like the tents and donkeys outside. Amazing.

**Marisha**: Look, Taliesin, itś a bardo. Oh my god!

**Taliesin**: No, go for it. Oh boy.

**Matt**: All right, so. 20 to 15?

**Laura**: 21.

**Matt**: All right.

**Laura**: Oh, wait, 22.

**Matt**: Nice. 15 to ten?

**Taliesin**: Ten.

**Sam**: Ten.

**Travis**: 12.

**Ashley**: 13.

**Liam**: 11.

**Matt**: So 13 and then ten and ten? Sorry, 12. So Yasha got 13, then we have Fjord. And then 11. And then we have Nott at ten. And whatd́ you get?

**Marisha**: Five. Rolled terribly.

Figure 10: A complex discussion of dice rolls. While many numbers are mentioned, there are actually seven rolls discussed in this segment: one initiative value for each player.

**Matt**: 16 hits. Go ahead and roll damage.

**Laura**: Yay. 4d6.

**Matt**: Yasha, youŕe almost on deck.

**Laura**: Ten, 16, 18, and then any attacks in the future– the next attack has advantage.

**Matt**: How much damage was that? 2d6, 18?

**Sam**: 4d6, you said.

**Laura**: Yeah, it was 4d6 radiant damage. So yeah, it was 18. And then Im̀ going to run the opposite direction away from him, but Im̀ going to keep my duplicate up there.

**Matt**: Okay, which, for your duplicate, I should go ahead and grab something.

Figure 11: A complex discussion of a damage roll calculation. Many numbers are floated in this discussion; the model must recognize that the discussion refers to a single damage event and the final, resolved damage was 18.

**Sam**: I thought you just said I had advantage!
**Matt**: You had advantage, but you have disadvantage because they're beyond the–
**Liam**: It cancelled out your advantage. It's just a straight roll and you lost the sneak attack.
**Sam**: Could I run forward and still get the sneak attack?
**Matt**: You could try it.
**Sam**: That's what I will do. Undo.
**Laura**: Roll again, because you might get natural 20!
**Sam**: Now I have to roll for stealth, right?
**Matt**: You've been stealthed as you crept up on this from the last time, so go for it.
**Sam**: I'm rolling to hit?
**Matt**: You already hit on that.
**Laura**: Roll again for advantage, you might get a 20.
**Sam**: Okay. Still 17 plus six.
**Matt**: Now you get a sneak attack, go for it.

Figure 12: This discussion highlights an *undo* of a dice roll. To tackle, the models needs to understand the narrative around the original roll and correct its count because of the undo.

