# OpenReview forum: "Oolong: Evaluating Long Context Reasoning and Aggregation Capabilities"
_ICLR.cc/2026/Conference — Submitted to ICLR 2026_

### Official Review · Reviewer_Nqhh · 2025-10-23

**Soundness:** 3
**Presentation:** 2
**Contribution:** 2
**Rating:** 2
**Confidence:** 4

**Summary:**

The paper introduces OOLONG, a long‑context benchmark aimed at measuring information aggregation rather than pure retrieval. It has two parts: OOLONG‑synth, which composes many short, easy ICL classification snippets into long contexts and asks distributional/user/timeline questions; and OOLONG‑real, which builds questions over transcripts from Critical Role with per‑episode statistics as gold labels. They find that strong models degrade markedly as context grows; even the best model tested struggles on both splits. The paper also provides ablations and a brief analysis of “reasoning effort” controls.

**Strengths:**

- Two splits. OOLONG‑synth allows controlled ablations; OOLONG‑real grounds the task in messy conversational data with externally curated labels. This pairing makes the claims more convincing than using only synthetic data.
- Performance vs. context length shows consistent degradation.
- “Reasoning level” comparisons are informative: more “reasoning effort” helps only at short lengths.

**Weaknesses:**

1. **Missing strong iterative/pipeline baselines.** - The paper does not report any strong baseline for tackling long-context problems (e.g. map‑reduce style baseline, an oracle‑labels pipeline upper bound, or a retrieval‑plus‑program). Including these would show how much of the gap stems from a single pass of LLMs vs. the fundamental difficulty of the task.

2. **OOLONG‑real data leakage.** - The gold labels come from CritRoleStats, a public source which have the possibility of being scraped for pre-training LLMs. Any efforts to check how much data leakage is present (e.g., just trying to ask the question without the context) can shed some more light on this.

3. **Limited Novelty.** - Several papers ([1,2]) have benchmarked LLM capabilities over long contexts that go beyond needle-in-the-haystack and measure the reasoning capabilities over the long context. Comparison against such benchmarks might make the paper's position clearer.

[1] Wang, Cunxiang, et al. "Novelqa: A benchmark for long-range novel question answering." CoRR (2024).
[2] Shaham, Uri, et al. "Scrolls: Standardized comparison over long language sequences." arXiv preprint arXiv:2201.03533 (2022).
[3] Bai, Yushi, et al. "Longbench v2: Towards deeper understanding and reasoning on realistic long-context multitasks." arXiv preprint arXiv:2412.15204 (2024).

**Questions:**

Several typos:
1. “WWe believe…”, line 100
2. “cummulative”, multiple times
3. Table 2 (instead of Table 1) in line 151

---

> ### Author Response · Authors · 2025-11-22
>
> Thank you for your review! We are glad to hear that you like the two-split structure of Oolong and find the reasoning level ablations informative. We address your concerns below.
>
> > Missing strong iterative/pipeline baselines. - The paper does not report any strong baseline for tackling long-context problems (e.g. map‑reduce style baseline, an oracle‑labels pipeline upper bound, or a retrieval‑plus‑program). Including these would show how much of the gap stems from a single pass of LLMs vs. the fundamental difficulty of the task.
>
> Thanks for your suggestion. We tested an iterative RAG baseline that uses GPT-5-nano as a code agent augmented with a BM25 retriever tool. As our retrieval unit, we used individual examples (Oolong-synth) and episode transcripts (Oolong-real). We evaluated this system on a 128k context window for Oolong-synth and up to two episode transcripts for Oolong-real. On a single inference run, we saw scores of 0.18 (Oolong-synth) and 0.23 (Oolong-real). We will run our iterative RAG baseline multiple times and report the average scores in the updated version of our paper. In our analysis, we find that RAG struggles on Oolong for a few reasons: (1) for Oolong-synth, the vast majority of input utterances are necessary to resolve any task (which we now more concretely analyze in the paper), and the RAG setup struggled at labeling individual examples before aggregation; (2) for Oolong-real, even when metadata on episode and utterance number is provided, reasoning over relative location of retrieved events in the transcript is challenging. Therefore, we tried full episodes as the retrieval unit, but the model still struggles at multi-episode questions that require richer information aggregation.
>
> > OOLONG‑real data leakage. - The gold labels come from CritRoleStats, a public source which have the possibility of being scraped for pre-training LLMs. Any efforts to check how much data leakage is present (e.g., just trying to ask the question without the context) can shed some more light on this.
>
> While the labels are derived from data on CritRoleStats, the Oolong questions take a very different form from raw CritRoleStats data– e.g. an “open-book” lookup for “what is the most common roll type over these episodes?” Using CritRoleStats would require mapping episode text to episode number, finding the spreadsheet on CritRoleStats that lists every roll in every episode, and then taking the maximum frequency roll over that subset of the data. Even if a model were trained on the data on CritRoleStats, memorization of that data alone would not be sufficient to answer Oolong questions.
>
> > Limited Novelty. - Several papers ([1,2]) have benchmarked LLM capabilities over long contexts that go beyond needle-in-the-haystack and measure the reasoning capabilities over the long context. Comparison against such benchmarks might make the paper's position clearer.
>
> We believe Oolong is sufficiently distinct from other benchmarks because of its combination of capability measurements: measuring both numeracy-based and reasoning-based aggregation in an easy-to-evaluate way, with naturalistic contexts and questions that often require the full input to answer correctly. We have added additional discussion of the differences between Oolong and existing long context benchmarks in the introduction and related work sections (Section 5).
>
> We would like to highlight that Oolong differs dramatically from both NovelQA and Scrolls: NovelQA can be solved largely through retrieval, and has potential data contamination issues due to its reliance on novels from Project Gutenberg for a large fraction of questions. Scrolls has many tasks which are challenging to evaluate and may suffer from contamination from pretraining (e.g. GovReport, where the gold summaries are taken from publicly available government reports. Scrolls was also meant for evaluating models finetuned on each subtask; the followup ZeroScrolls, which we discuss in the text, has one task that is similar to one of the simpler tasks in Oolong but largely has the same issues as Scrolls.
>
> > typos
>
> Thank you, we have corrected these!
>
> Thank you again for the feedback; we are happy to discuss any points further.

---

### Official Review · Reviewer_S39y · 2025-10-29

**Soundness:** 3
**Presentation:** 3
**Contribution:** 1
**Rating:** 2
**Confidence:** 4

**Summary:**

The paper introduces OOLONG, a benchmark for evaluating long-context reasoning in LLMs. It is split into two parts:

OOLONG-synth, which builds very long prompts by concatenating many short, labeled classification examples (spam detection, sentiment, etc.) and then asks distributional / aggregation questions like “how many positive reviews?” or “which label is most frequent?”, including temporal and per-user variants.

OOLONG-real, which uses long, noisy, real conversational transcripts from Critical Role (Dungeons & Dragons actual play) and asks questions about events and statistics across one or more episodes (“how many spells of type X were cast?”, “what’s the cumulative roll count by episode N?”).

Across a range of large models (GPT-5, Claude-Sonnet-4, DeepSeek R1, etc.), accuracy drops sharply as context grows, and no model exceeds 50% accuracy at 128K tokens on either split. The authors argue this shows that current LLMs still struggle not just with retrieval (“find the one relevant span”) but with aggregating many small local decisions across extremely long inputs.

**Strengths:**

1. OOLONG frames long-context reasoning as multi-step aggregation: identify relevant spans, classify locally, and pool globally (counts, timelines, user-specific patterns). This is positioned as closer to realistic analytics tasks than classic “needle in a haystack.”

2. OOLONG-synth is controllable: it uses standard classification datasets (spam, sentiment, NLI, etc.) and scales to millions of tokens, letting the authors ablate factors like context length, label access, and time/user structure.

3. OOLONG-real uses long, messy human dialogue from Critical Role, combined with fan-annotated gold stats (dice rolls, spell usage), to approximate “real-world” multi-episode log analysis.

4. The paper confirms a now-familiar story: models nominally supporting 100K+ tokens still degrade badly when asked to actually integrate information across that full window, especially beyond ~64K. Even top models sit <50% at 128K.

5. They dig into DeepSeek R1: on OOLONG-real it can behave competitively, but on OOLONG-synth it underperforms even the random baseline because it burns tokens “thinking out loud,” tries to label every item, and then times out before answering. That illustrates the bottleneck they care about: planning and aggregation under length pressure.

**Weaknesses:**

My main issue is novelty. I did not find a clear, convincing argument that OOLONG measures a fundamentally new capability beyond what recent long-context benchmarks and analyses already target.

1. The stated motivation is: most long-context tests are retrieval-style (needle-in-a-haystack, MRCR, etc.), whereas OOLONG requires aggregation/counting over many items. But this does not feel substantially different from what earlier work like BABILong / long multi-step reasoning tasks including object tracking and counting (and similar multi-supporting-fact QA) that explicitly require pooling facts from many snippets across long input. RULER / HELMET-style setups already do with multi-hop tracing, counting, and distributional questions across long synthetic inputs, where relevant bits are dispersed and must be combined rather than simply located. The paper itself cites RULER, HELMET, MRCR, ZeroSCROLLS, and related aggregation-style tasks such as estimating label distributions in reviews.

The paper positions OOLONG as fundamentally new (“to the best of our knowledge, there are no benchmarks that evaluate LM’s ability to perform information aggregation at scale”). I don’t think that claim is fully justified. The differences feel incremental (different source data, different instructions), not qualitatively new.

2. Findings largely reproduce existing conclusions. The high-level conclusions are not surprising relative to prior long-context studies, and in fact mostly restate them:
- Performance falls off quickly with longer contexts.
- Even frontier models under 200K+ context struggle to actually use that context.
- Asking for “more reasoning” doesn’t magically fix degradation at very long lengths.
- Some models try brute-force per-chunk labeling, run out of space, and never answer.

These are all consistent with what benchmarks like BABILong/RULER/HELMET already report: long-context models often fail not because they can’t read long input but because they can’t maintain and aggregate distributed evidence across it. The paper presents similar failure patterns, just in a new wrapper.


Overall, I agree that long-context aggregation is important. I am less convinced that OOLONG, as currently framed, is novel enough beyond BABILong-/RULER-style counting & aggregation tasks and MRCR-/narrative-memory-style multi-session tracking. The empirical story (“even SOTA models still can’t do it at 128K+ tokens”) mostly confirms what those benchmarks already argued.

**Questions:**

1. Can you articulate what OOLONG measures that BABILong/RULER/HELMET/MRCR-style multi-hop aggregation does not already measure? Point to a specific capability that would let a model ace BABILong/RULER but fail OOLONG.

2. For OOLONG-real, can you show that Critical Role transcripts introduce qualitatively new reasoning structure (e.g. cross-episode state changes, retcons, overlapping threads) that existing narrative / long-dialogue benchmarks don’t already capture? Right now it reads mostly like very long multi-session dialogue logs plus counting.

3. Is there any system-level insight that changes how we evaluate or train long-context models because of OOLONG, beyond “they still struggle past 64K”?

---

> ### Author Response · Authors · 2025-11-22
>
> Thank you for your review! We are glad you agree that information aggregation is an important task, and see merits in Oolong-synth’s controllability and Oolong-real’s more realistic data. We address your questions below.
>
> > this does not feel substantially different from what earlier work like BABILong / long multi-step reasoning tasks including object tracking and counting (and similar multi-supporting-fact QA) that explicitly require pooling facts from many snippets across long input. RULER / HELMET-style setups already do with multi-hop tracing, counting, and distributional questions across long synthetic inputs, where relevant bits are dispersed and must be combined rather than simply located.
>
> > (“to the best of our knowledge, there are no benchmarks that evaluate LM’s ability to perform information aggregation at scale”). I don’t think that claim is fully justified.
>
> We’ve removed this language and added more fine-grained discussion of differences from existing benchmarks instead; see the revised introduction and related work (Section 5, Table 3). We briefly summarize here as well.
>
> We contrast Oolong with MRCR, BABILong, RULER and HELMET (includes InfinityBench) on various capabilities: real vs synthetic data, aggregation capabilities and measurability. We believe these capabilities in combination provide an ideal testbed for long-context LLMs. As we show in Table 3, Oolong (synth + real) is the only dataset that covers all of these capabilities. While RULER includes certain aggregation capabilities, it solely relies on synthetic data. BABILong covers multi-step reasoning and realistic questions, but it fully relies on unrelated distractor text and doesn't necessarily require the full input to answer the question. HELMET (LongQA and Summarization) includes real dataset and aggregation, but lacks numeracy questions. Additionally, summarization-based tasks present challenges in automatic evaluation.
>
> > The high-level conclusions are not surprising relative to prior long-context studies
>
> While it’s true that most long context benchmarks show performance drops with increasing context length, we want to highlight that Oolong presents tasks that resemble real-world use of long-context models. As we mentioned above (Table 3), Oolong includes realistic questions, and distractor texts that match with relevant text. The reasoning aspects of Oolong are novel.
>
> > Point to a specific capability that would let a model ace BABILong/RULER but fail OOLONG.
>
> RULER’s aggregation tasks are heavily counting-based; if you could perform counting but no other type of reasoning (e.g. if you could call python’s itertools.Counter()), you could solve these tasks.
> BABILong requires identifying the relevant sentences from a haystack and then reasoning. If you could do perfect reasoning at <1000 token scale and perfect identification of relevant sentences (RULER-style), you could solve BABILong by pulling the relevant sentences out and then treating this as a short-context task (this reduces it to the original BABI task, which modern models can generally do quite well). But you can’t do that with Oolong; Oolong-synth is far more info-dense, so you would need to reason at >1000 token scale, and Oolong-real requires complicated and varying amounts of context to resolve each reasoning step (see response to next point). Oolong-real resembles a needlestack, not a haystack.

---

> > ### Author Response · Authors · 2025-11-22
> >
> > > can you show that Critical Role transcripts introduce qualitatively new reasoning structure (e.g. cross-episode state changes, retcons, overlapping threads) that existing narrative / long-dialogue benchmarks don’t already capture?
> >
> > Thanks for highlighting this! You’ve already identified some of the differences [e.g. the retcons and state changes].  We’ve added an appendix (Appendix E) that highlights some of the unique challenges of the Critical Role data, which we briefly summarize below:
> >
> > 1. Character-actor resolution
> > 2. In-character vs out-of-character / joking
> > 3. Overlapping dialogue
> > 4. retcon/reversal
> > 5. Varying-length discussion for a single role value, including recalculation
> >
> > > Is there any system-level insight that changes how we evaluate or train long-context models because of OOLONG, beyond “they still struggle past 64K”?
> >
> > The low performance on Oolong suggests more work is necessary in developing both reasoning abilities and more general long context abilities. Even in short context regimes (8K tokens or less), strong models cannot perfectly perform the multi-step reasoning tasks in Oolong-synth. However, in longer context regimes, many models perform roughly equivalently on Oolong-synth and Oolong-real at the same context length, despite the much higher information density in Oolong-synth. This suggests that better length generalization, regardless of reasoning ability, may also improve performance on this type of challenging long context aggregation task.
> > However, some models behave differently in extremely information-dense regimes. In particular, both Gemini-2.5-Pro and Deepseek-R1-0528 are strong models that perform well on Oolong-real but fail in Oolong-synth because of an over-reasoning phenomenon. This highlights a direction for future work. While prior work on reasoning chains has focused on over-reasoning for adversarial or overly simple problems, Oolong indicates that even strong models may struggle to plan the reasoning quantity for information-dense inputs, where it may be desirable to accept a less optimal or more error-prone reasoning strategy (e.g. double-checking less frequently) in order to avoid running into the maximum reasoning tokens. Future models that are aware of the maximum reasoning token budget may be able to more carefully plan reasoning strategies according to the allowed budget.
> >
> > We updated the conclusion section with this explanation.
> >
> > Thank you for the thoughtful questions. We are happy to discuss further.

---

> > ### Comment · Reviewer_S39y · 2025-11-26
> > **Not yet convincedd**
> >
> > I would like to thank authors for providing answers. Unfortunately, I'm not convinced that contribution of this work is significant enough to be accepted. Novelty compared to existing benchmarks is limited, and no novel insights about LLM capabilities are presented.

---

> > > ### Author Response · Authors · 2025-11-30
> > >
> > > We respectfully disagree-- while there have been long-context benchmarks evaluating reasoning in synthetic settings (including Babilong), Oolong-real's naturalistic dialogue setting, coupled with the ease of ablating components of the task in Oolong-synth, allows us to identify several novel insights. Oolong shows that:
> > > * models struggle at specifically the aggregation step of multi-step reasoning
> > > * temporal reasoning is particularly challenging, even for models that perform very well at reasoning tasks at shorter context
> > > * some models (e.g. Gemini 2.5 and Deepseek R1) show diverging patterns of performance depending on how information-dense the task is, even at the same context length, because of inefficient use of reasoning tokens
> > > * for models that do not suffer from this pathology of reasoning, the difficulty of a task correlates more with context length than with the density of information necessary to solve the task
> > >
> > > These last two insights are particularly surprising and point to directions to further improve modeling for long context reasoning.
> > >
> > > While we recognize that the reviewer cannot reply further because ICLR has now restricted reviewer commenting, we wish to highlight these points to the AC.

---

### Official Review · Reviewer_V1ms · 2025-11-01

**Soundness:** 2
**Presentation:** 3
**Contribution:** 2
**Rating:** 4
**Confidence:** 4

**Summary:**

Oolong is a long-context reasoning benchmark with two sets of tasks. One set uses existing long-context datasets as subtasks and requires answering 3 types of composite questions based on them. Another set uses long conversational data and formulates 3 types of questions based on counting, enumeration and indexing of events in the data. The experiments show that the tasks are challenging for frontier long-context models, with performance significantly decreasing with context size.

**Strengths:**

- Oolong provides two challenging sets of tasks that underscore the limitations of LLMs in long-context scenarios.

- The sample size is scalable, while context usage remains more dense compared to classic needle-in-a-haystack - based benchmarks. Relevant facts are hard to distinguish from irrelevant ones, which makes the tasks challenging for LLMs.
- Following relevant works, Oolong-synth inherits the validation-test structure, with no overlap between them, which allows for more fair evaluation.

**Weaknesses:**

- Conceptually the contribution of the current work is limited, as LLM degradation in long-context scenarios has been demonstrated by multiple other synthetic and real-world benchmarks. Some of these works were listed by authors in Section 5, but several relevant works that require long-context reasoning with aggregation are missing, including BABILong (Kuratov et al., 2024) and InfinityBench (Zhang et al., 2024). Discussing their differences and limitations compared to Oolong is advised.
- If the primary contribution of the work is the study of aggregation capabilities, a more in-depth analysis is needed, not only reported degradation with length.
- The task complexity of the Oolong-synth set increases with context size, making it hard to estimate the effect of these two matters independently. Generally speaking, it is not entirely clear whether the performance degrades solely because of the increasing context size or because the underlying aggregation subtasks become more complex. Evaluating retrieval-augmented models or multi-step prompting would provide some insight in this regard, however no such baselines are present.
- The evaluation of certain models such as Deepseek R1 might be not completely fair. As authors state in section 4.3, the results are inconsistent, and traces often end mid-sentence, without outputting any answer. In line 201 it is confirmed that some models ran out of output tokens before providing the answer. Instead of evaluating the reasoning quality, this way of inference measures the performance under the CoT length limitation. It is not necessarily wrong, but this can cause confusion and misinterpretation of the results.

**Questions:**

- Oolong-synth samples are constructed by mixing contexts of various long-context tasks. One concern is related to information overlapping between these contexts: can one task affect the solution of another one? If so, what can be done to prevent this effect?
- The types of questions in Oolong-real can be more suitable for code executing models, they can have a significant upper hand. Have you tried using such models in the evaluation?
- Oolong-real has really long samples of length 1.3M, and Oolong-synth up to 4M tokens,  but were they actually used? Max length reported in Table 4 is 175K and 512K on Figures 2, 3, 4.


comments / typos:
- Appendix B3: results for all models are mentioned but not provided
- Line 96: missing space
- Lines 100-101, typo: We believe

---

> ### Author Response · Authors · 2025-11-22
>
> Thank you for your review! We are glad you find Oolong to be challenging and scalable. We address each concern in turn below.
>
> > Conceptually the contribution of the current work is limited, as LLM degradation in long-context scenarios has been demonstrated by multiple other synthetic and real-world benchmarks. Some of these works were listed by authors in Section 5, but several relevant works that require long-context reasoning with aggregation are missing, including BABILong (Kuratov et al., 2024) and InfinityBench (Zhang et al., 2024). Discussing their differences and limitations compared to Oolong is advised.
>
> Thank you for the references! We have added more discussion of these and related works in the paper to highlight the difference with Oolong (section 5, Table 3). We briefly summarize here as well.
>
> We contrast Oolong with MRCR, BABILong, RULER and HELMET (includes InfinityBench) on various capabilities: real vs synthetic data, aggregation capabilities and measurability. We believe these capabilities in combination provide an ideal testbed for long-context LLMs. As we show in Table 3, Oolong (synth + real) is the only dataset that covers all of these capabilities. While RULER includes certain aggregation capabilities, it solely relies on synthetic data. BABILong covers multi-step reasoning and realistic questions, but it fully relies on unrelated distractor text and doesn't necessarily require the full input to answer the question. HELMET (LongQA and Summarization) includes real dataset and aggregation, but lacks numeracy questions. Additionally, summarization-based tasks present challenges in automatic evaluation.
>
> > If the primary contribution of the work is the study of aggregation capabilities, a more in-depth analysis is needed, not only reported degradation with length.
>
> Thank you for the suggestion! We have added more analysis in Section 4.2 (splits by task type and answer type) and Section 4.3 (more reasoning analysis).
>
> In 4.2, we measure model performance across task and answer types, and multiple context window lengths. We observe that temporal questions are the most challenging for the models. Questions that require a date or month/year (e.g. ``January 2021'') as an answer show generally lower performance for the same model, and show greater spread in model capabilities than the other answer categories. For instance, the gap between GPT-5 and GPT-5-nano performance is more than 4x larger for questions that require outputting a date than for questions that require outputting a label. Relative model performance is mostly stable across answer types, although Claude-Sonnet-4 is relatively much stronger on numerical reasoning and comparisons than the GPT series models. On Oolong-real, we found frequency-based questions to be the hardest. Gemini-2.5-Pro and GPT-5 do reasonably well on enumeration and indexing while other models consistently struggle at these questions.
>
> We also expanded section 4.3 to include additional analysis of reasoning traces for DeepSeek R1 and Gemini 2.5 Pro. DeepSeek R1 particularly struggled with information-dense Oolong-synth, and its strategy to label individual examples before deciding on relevant examples led it to run out of reasoning tokens. We also benchmarked Gemini 2.5 Pro, and found that Gemini 2.5 Pro struggles with reasoning for long inputs. This behavior is more pronounced in the information-dense Oolong-synth.
>
> > The task complexity of the Oolong-synth set increases with context size, making it hard to estimate the effect of these two matters independently. Generally speaking, it is not entirely clear whether the performance degrades solely because of the increasing context size or because the underlying aggregation subtasks become more complex.
>
> This is an interesting point. For Oolong, we believe that the length is an important part of the aggregation task. Prior work (Levy et al., 2024) has looked at tasks that disentangle length from reasoning. However, their analysis was done by incrementally adding irrelevant text drawn from other sources. As we highlight in our Table 3 (new), Oolong uses distractors that match the relevant text. This follows recent benchmarks such as MRCR and HELMET.
> Levy et al., 2024. Same Task, More Tokens: the Impact of Input Length on the Reasoning Performance of Large Language Models. ACL.

---

> > ### Author Response · Authors · 2025-11-22
> >
> > > Evaluating retrieval-augmented models or multi-step prompting would provide some insight in this regard, however no such baselines are present.
> >
> > Thanks for your suggestion. We tested an iterative RAG baseline that uses GPT-5-nano as a code agent augmented with a BM25 retriever tool. As our retrieval unit, we used individual examples (Oolong-synth) and episode transcripts (Oolong-real). We evaluated this system on a 128k context window for Oolong-synth and up to two episode transcripts for Oolong-real. On a single inference run, we saw scores of 0.18 (Oolong-synth) and 0.23 (Oolong-real). We will run our iterative RAG baseline multiple times and report the average scores in the updated version of our paper. In our analysis, we find that RAG struggles on Oolong for a few reasons: (1) for Oolong-synth, the vast majority of input utterances are necessary to resolve any task (which we now more concretely analyze in the paper), and the RAG setup struggled at labeling individual examples before aggregation; (2) for Oolong-real, even when metadata on episode and utterance number is provided, reasoning over relative location of retrieved events in the transcript is challenging. Therefore, we tried full episodes as the retrieval unit, but the model still struggles at multi-episode questions that require richer information aggregation.
> >
> > > The evaluation of certain models such as Deepseek R1 might be not completely fair. As authors state in section 4.3, the results are inconsistent, and traces often end mid-sentence, without outputting any answer. In line 201 it is confirmed that some models ran out of output tokens before providing the answer. Instead of evaluating the reasoning quality, this way of inference measures the performance under the CoT length limitation. It is not necessarily wrong, but this can cause confusion and misinterpretation of the results.
> >
> > We agree with the reviewer that reasoning quality is important. In fact, we think reasoning quality includes not over reasoning. We show this limitation of R1 through our diagnosis on Oolong-synth. We also expanded Section 4.3 to include more detailed diagnosis of DeepSeek R1 performance.
> >
> > > Oolong-synth samples are constructed by mixing contexts of various long-context tasks. One concern is related to information overlapping between these contexts: can one task affect the solution of another one? If so, what can be done to prevent this effect?
> >
> > While Oolong-synth uses data from 10 different ICL datasets, each individual Oolong-synth sample draws data from a single ICL dataset. Thus, information overlapping across the ICL tasks should not be an issue.
> >
> > > The types of questions in Oolong-real can be more suitable for code executing models, they can have a significant upper hand. Have you tried using such models in the evaluation?
> >
> > Thank you for the interesting suggestion! Do you have a recommendation of a code executing model you would like to see evaluated? We would be happy to add it.
> >
> > > Oolong-real has really long samples of length 1.3M, and Oolong-synth up to 4M tokens, but were they actually used? Max length reported in Table 4 is 175K and 512K on Figures 2, 3, 4.
> >
> > You're correct that we did not use the longest examples– no model tested supported these context lengths. We see this as future-proofing for the next generations of longer context models.
> >
> > > typos
> >
> > Thank you! We have corrected these.
> >
> > > Appendix B3: results for all models are mentioned but not provided
> >
> > Apologies for the error; we moved this to the main text (Table 4, which has now been moved back into the appendix) and left the reference behind.
> >
> > Thank you again for the feedback; we are happy to discuss any points further.

---

### Official Review · Reviewer_p6wJ · 2025-11-01

**Soundness:** 2
**Presentation:** 3
**Contribution:** 3
**Rating:** 4
**Confidence:** 3

**Summary:**

The paper introduces Oolong, a benchmark for long-context reasoning and aggregation that goes beyond simple retrieval. Oolong comprises two splits: Oolong-synth and Oolong-real (Dungeons & Dragons campaign transcripts). Tasks require analyzing atomic chunks and aggregating the results (classification, counting, temporal/user relations). Experiments show that even frontier models perform poorly, with accuracy dropping below 50% already at 128K context length. The authors release data and code to facilitate further work on models that can reliably aggregate information over long contexts.

**Strengths:**

* Practical Significance: Proposed benchmark tries to move beyond common needle-in-a-haystack or generic summarization setups, targeting fine-grained analysis and aggregation. These areas underrepresented in existing long-context benchmarks.
* Originality: Requires atomic reasoning, counting/classification, and temporal/user-relational aggregation rather than simple retrieval, reducing shortcut solutions.
* Realistic long-horizon data: Uses Dungeons & Dragons campaign transcripts, which feature coherent narratives, long-range dependencies, and non-encyclopedic facts less likely to be memorized from pretraining.

**Weaknesses:**

* Limited experimental validation of claims: The paper argues Oolong differs from retrieval-centric long-context benchmarks by requiring multi-chunk analysis and logical aggregation, but provides no targeted experiments to substantiate this. To demonstrate the distinction empirically, the evaluation should include not only base LLMs but also strong single- and multi-step RAG baselines. Divergent performance of such RAG systems on Oolong vs. needle-in-a-haystack / multi-hop retrieval tasks would directly support the benchmark’s claimed novelty.

* Insufficient result granularity: Results are reported mainly by model and average sample length, with no breakdown by task or question type. A per-category analysis (for example, classification, counting, and temporal or user-relational queries) is needed to reveal inherent difficulty and interactions with context length. Some categories may degrade disproportionately at longer contexts. Inclusion of plots or tables that slice performance by task type and by length would strengthen the paper.

**Questions:**

* How was the evaluation metric chosen in lines 203–208, and will you report additional metrics?

* Real data quality. The transcripts are probably auto-generated from hundreds of 4 hour long streams and hard to verify manually, and the website-sourced data may include errors without accuracy guarantees. Can you more thoroughly describe your methods for assessing data quality and quantify how transcription or source errors affect model scores on Oolong-real?

* Can parts of data cleaning and annotation be automated, for example with LLM-based pipelines?

---

> ### Author Response · Authors · 2025-11-22
>
> Thank you for your review! We are glad to hear that you find Oolong practically significant and agree that the real split is realistic long-horizon data. We address your points below.
>
> > the evaluation should include not only base LLMs but also strong single- and multi-step RAG baselines.
>
> Thanks for your suggestion. We tested an iterative RAG baseline that uses GPT-5-nano as a code agent augmented with a BM25 retriever tool. As our retrieval unit, we used individual examples (Oolong-synth) and episode transcripts (Oolong-real). We evaluated this system on a 128k context window for Oolong-synth and up to two episode transcripts for Oolong-real. On a single inference run, we saw scores of 0.18 (Oolong-synth) and 0.23 (Oolong-real). We will run our iterative RAG baseline multiple times and report the average scores in the updated version of our paper. In our analysis, we find that RAG struggles on Oolong for a few reasons: (1) for Oolong-synth, the vast majority of input utterances are necessary to resolve any task (which we now more concretely analyze in the paper), and the RAG setup struggled at labeling individual examples before aggregation; (2) for Oolong-real, even when metadata on episode and utterance number is provided, reasoning over relative location of retrieved events in the transcript is challenging. Therefore, we tried full episodes as the retrieval unit, but the model still struggles at multi-episode questions that require richer information aggregation.
>
> > A per-category analysis (for example, classification, counting, and temporal or user-relational queries) is needed to reveal inherent difficulty and interactions with context length. Some categories may degrade disproportionately at longer contexts. Inclusion of plots or tables that slice performance by task type and by length would strengthen the paper.
>
> Thank you for the suggestion! We have included this analysis in section 4.2 and appendix D. We observe that temporal questions are the most challenging for the models. Questions that require a date or month/year (e.g. ``January 2021'') as an answer show generally lower performance for the same model, and show greater spread in model capabilities than the other answer categories. For instance, the gap between GPT-5 and GPT-5-nano performance is more than 4x larger for questions that require outputting a date than for questions that require outputting a label. Relative model performance is mostly stable across answer types, although Claude-Sonnet-4 is relatively much stronger on numerical reasoning and comparisons than the GPT series models. On Oolong-real, we found frequency-based questions to be the hardest. Gemini-2.5-Pro and GPT-5 do reasonably well on enumeration and indexing while other models consistently struggle at these questions.

---

> > ### Author Response · Authors · 2025-11-22
> >
> > > How was the evaluation metric chosen in lines 203–208, and will you report additional metrics?
> >
> > The evaluation metric (exponentially diminishing score with distance from correct answer) was chosen as a soft backoff to allow partial credit for numerical answers that are close (e.g. the model miscounts the number of examples in a class by 1). This is only for numeric response types; every other response type is scored on 0/1 accuracy. If there are other metrics you think would be appropriate for this task, we would be happy to run them and compare!
> >
> > > Real data quality. The transcripts are probably auto-generated from hundreds of 4 hour long streams and hard to verify manually, and the website-sourced data may include errors without accuracy guarantees. Can you more thoroughly describe your methods for assessing data quality and quantify how transcription or source errors affect model scores on Oolong-real?
> >
> > Thank you for this question! The transcripts in question are *not* auto-generated – they were first hand-transcribed by members of the fan community, who implemented a multi-step editing process with editorial guidelines. These were then further cleaned by Rameshkumar and Bailey (2020) to resolve any remaining inconsistencies. The stats were manually compiled by volunteers and the team at CritRoleStats. While it’s not possible to completely eliminate sources of potential error, these types of fan-documentation processes are fastidiously annotated and cross-checked by multiple people– a standard of care far higher than that of a paid annotator with no connection to the task or data domain. We’ve added more detail about the data annotation processes in Appendix E.
> >
> > > Can parts of data cleaning and annotation be automated, for example with LLM-based pipelines?
> >
> > Our verification step for ICL examples (section 2.1) is automated using LLMs! The Oolong-real data in the paper was cleaned and annotated by fan projects, and represents a human domain-expert gold standard. Trying to automate cleaning/annotation for this kind of data would be an interesting challenge, but one that's outside the scope of this work.
> >
> > Thank you again for the feedback; we are happy to discuss any points further.

---

### Author Response · Authors · 2025-11-22

To address reviewer comments, we have added substantial changes in the following sections. These sections are also highlighted in blue in the text of the updated paper, for easy identification.

- Introduction: more clear framing of the difference from existing long-context reasoning works in the first two paragraphs.
- Related work: we have added more discussion in the long-context benchmarks section, as well as a new table (Table 3) to make it clearer how Oolong differs from existing benchmarks.
- Fine-grained analysis: we add more analysis by task type and varying over length; this appears in Section 4.2 and Appendix D.
- Oolong-real: we discuss data quality, curation and challenges unique to Oolong-real in Appendix E.
- Moving content to Appendix: to make space for the additional analysis, we moved the leaderboard and a few details about data validation to the Appendix.
- Conclusion: more actionable insights and discussion of future work, in the middle two paragraphs.

---

### Meta-Review · Area_Chair_QJFz · 2025-12-28

**Summary:**

Paper Summary. This paper introduces a benchmark Oolong for long-context reasoning and aggregation. It requires analyzing individual text chunks and then logical aggregation of analyses to answer questions. It  includes tasks of classification, counting, temporal/user relations. Experiments show that frontier models perform poorly on Oolong, and the best model GPT-5 only achieves below 50% accuracy for context of 128K.

Paper Strengths. (1) The papers targets an important issue of context reasoning and aggregation, moving beyond needle-in-a-haystack or generic summarization capabilities. (2) The dataset contains both controllable synthetic data split and realistic data split of long-horizon  dialog with coherent narratives and long-range dependencies.

Reviewer Concerns. (1) There is limited experiments to substantiate the claim that Oolong requires multi-chunk analysis and logical aggregation and differs from existing retrieval-centric benchmarks. It is not clear whether the performance drops stems from solely the longer context size or the more complex underlying aggregation subtasks. (Reviewer p6wJ, Reviewer V1ms, Reviewer Nqhh) (2) Experiment results shall provide more fine-grained performance analysis with breakdown by task, question type, and context length  (Reviewer p6wJ, Reviewer V1ms). (3) The conceptual novelty of contribution is limited compared to existing work. Especially, it is not convincing that Oolong measures a fundamentally new capability beyond existing long-context benchmarks such as BABILong/RULER/HELMET, and experimental findings are not new compared to existing conclusions (Reviewer V1ms, Reviewer S39y, Reviewer Nqhh) (4) The evaluation of certain models such as Deepseek R1 might be not completely fair and needs further discussion (Reviewer V1ms) (5) There is also a concern on  data leakage of the real split (Reviewer Nqhh).

**Reviewer Concerns:**

The following concerns are addressed

(1) Author rebuttal provides further experiments using iterative RAG on Oolong. The performance is low with accuracy of 0.18 (Oolong-synth) and 0.23 (Oolong-real), indicating the challenge of Oolong for RAG type methodology.
(2) Author rebuttal mentions that more fine-grained analysis is provided in Section 4 and Appendix.
(4) More discussion are provided regarding DeepSeek R1 overthinking issue.
(5) Author rebuttal clarifies that data leakage is not a major concern.

However, the following concerns are still outstanding.
(3) While author adds more discussion of related works in the updated paper, after checking the rebuttal, it is not entirely convincing that the novelty and contribution of this work is significant enough (Reviewer S39y).

Overall, I believe this paper presents a very interesting dataset for an important issue, yet it will need substantial revision to highlight the conceptual novelty and major new findings.

**Reviewer Scores:**

Based on the rebuttal and outstanding reviewer concerns, a very optimistic anticipation is that the score will change as follows

Reviewer p6wJ: 4 -> 6

Reviewer V1ms: 4 -> 6

Reviewer S39y: 2 -> 2 due to the outstanding concern (3)

Reviewer Nqhh: 2 -> 4

---

### Decision · Program_Chairs · 2026-01-26

Reject